# FAST PROTEOME-SCALE PROTEIN INTERACTION RETRIEVAL VIA RESIDUE-LEVEL FACTORIZATION

**Jianan Zhao[1,2], Zhihao Zhan[1,2], Narendra Chaudhary[3], Xinyu Yuan[1,2], Zuobai Zhang[1,2], Qian Cong[4], Jian Zhou[4], Sanchit Misra[3], Jian Tang[1,5,6]** [*]
[1]Mila - Québec AI Institute, [2]Université de Montréal, [3]Intel Corporation
[4]University of Texas Southwestern Medical Center, [5]HEC Montréal, [6]CIFAR AI Chair

## ABSTRACT

Protein-protein interactions (PPIs) are mediated at the residue level. Most sequence-based PPI models consider residue-residue interactions across two proteins, which can yield accurate interaction scores but are too slow to scale. At proteome scale, identifying candidate PPIs requires evaluating nearly *all possible protein pairs*. For $N$ proteins of average length $L$, exhaustive all-against-all search requires $\mathcal{O}(N^2 L^2)$ computation, rendering conventional approaches computationally impractical. We introduce RaftPPI, a scalable framework that approximates residue-level PPI modeling while enabling efficient large-scale retrieval. RaftPPI represents residue interactions with a Gaussian kernel, approximated efficiently via structured random Fourier features, and applies a low-rank factorized attention mechanism that admits pooling into a compact embedding per protein. Each protein is encoded once into an indexable embedding, allowing approximate nearest-neighbor search to replace exhaustive pairwise scoring, reducing proteome-wide retrieval from *months* to *minutes* on a single GPU or CPU. On the human proteome with the D-SCRIPT dataset, RaftPPI retrieves the top 20% pairs from ∼200M candidate pairs in 5.7 minutes on an A100 GPU, or 3.3 minutes on an Intel® Xeon® 6980P CPU, covering 75.1% of the true interacting pairs, compared to 4.9 GPU months for the best prior method (61.2%). Across seven benchmarks with sequence- and degree-controlled splits, RaftPPI achieves state-of-the-art PPI classification and retrieval performance, while enabling residue-aware, retrieval-friendly screening at proteome scale.

## 1 INTRODUCTION

Protein-protein interaction (PPI) is central to understanding cellular mechanisms and enabling applications in target discovery (Loscalzo, 2023), pathway reconstruction (Ritz et al., 2016), and functional annotation (Sharan et al., 2007). In practice, many discovery tasks require proteome-scale screening, scoring protein pairs within a species to surface plausible interactors (Humphreys et al., 2024; Zhang et al., 2024). However, proteome-scale PPI prediction remains time-consuming due to the quadratic number of candidate protein pairs. One high-accuracy route is to predict multimer structures with end-to-end structure predictors (Jumper et al., 2021; 2024; Evans et al., 2021). While accurate, these pipelines rely on MSAs/templates and require $\mathcal{O}(L^3)$ complexity to update pairwise representations of proteins with length $L$. This daunting cost makes per-complex structure prediction difficult to amortize across large candidate sets. An alternative frames PPI as binary classification from alignment-free PLM embeddings (Sledzieski et al., 2024; Ko et al., 2024; Liu et al., 2024). Although per-sequence encoding is $\mathcal{O}(L^2)$, these models must jointly encode each protein *pair* at inference; thus an exhaustive screen over the human proteome with ∼ 20,000 proteins entails $\approx 2 \times 10^8$ candidate pairs, making large-scale screening computationally prohibitive. To illustrate, the state-of-the-art PPI classification model PLM-Interact (Liu et al., 2024) requires 148.47 A100 GPU-days (≈4.9 months) to screen the human proteome (see Table 3 and § 4.3).

In light of these limitations, we propose **R**esidue-interaction **A**pproximation with **F**ourier Fea**T**ures (RaftPPI), which models PPI by *approximating residue-level interactions while enabling scalable protein retrieval*. RaftPPI models residue–residue scores with a Gaussian kernel and aggregates them

---

[*]Corresponding author. Code release: https://github.com/AndyJZhao/RaftPPI

to a protein-level score (Fig. 1). The non-linear kernel is efficiently approximated via random Fourier features (Rahimi and Recht, 2007; Yu et al., 2016), and pooling uses a low-rank factorized attention that admits a linear-time approximation at inference. As a result, each protein is encoded once into a fixed-length representation amenable to approximate nearest-neighbor search, e.g., Hierarchical navigable small world (HNSW; (Malkov and Yashunin, 2020)), to retrieve likely interactors—retaining residue-level interactions while avoiding explicit per-pair computation. In practice, the dominant cost is PLM encoding, giving overall $\mathcal{O}(NL^2)$ across a proteome; on a single A100 or a single Intel Xeon 6 CPU, retrieving the top 20% of human-proteome pairs completes in minutes, achieving a $10^4$-fold speedup over prior art.

Besides model design, we also seek to mitigate challenges from the lack of reliable negative data in PPI datasets. Experimentally confirmed non-interactions are rare, so negatives are typically *constructed*, and their quality varies widely (Neumann et al., 2022; Zhao et al., 2022). Random or compartment-based pairing often produces overly easy, biased examples that lead to overly optimistic results, whereas co-localized, functionally related, or topology-aware sampling yields harder and more informative ones (Ben-Hur and Noble, 2006; Park and Marcotte, 2011; 2012; Zhang et al., 2018). These observations motivate our *adaptive negative weighting* loss, which applies self-adversarial weights (Sun et al., 2019) based on model confidence so that harder negatives are assigned greater weights, mitigating biases from easy constructed pairs.

**Contributions.** We introduce RaftPPI, a retrieval-friendly, residue-aware framework that compresses each protein into an indexable embedding for efficient ANN search, reducing proteome-level PPI screening from *GPU months* to *minutes* on a single GPU or CPU. With an adaptive negative-weighting loss that emphasizes hard negatives and mitigates the lack of reliable negative data, RaftPPI achieves strong classification and retrieval performance under rigorous sequence- and degree-controlled benchmarks.

## 2 RELATED WORK

**Protein Language Models.** Transformer-based Protein Language Models (PLMs) pretrained on large sequence corpora (e.g., ProtTrans (Elnaggar et al., 2021), ESM 1b (Rives et al., 2021), and ESM 2 (Lin et al., 2023)) learn residue-level embeddings that implicitly capture evolutionary and structural priors. Generative PLMs such as ProGen and ProGen2 (Madani et al., 2020; Nijkamp et al., 2022) use autoregressive modeling for controllable sequence design. Beyond sequence-only vocabularies, Foldseek (van Kempen et al., 2024) introduces discrete 3D structure tokens that have been used to augment PLMs with structure-aware vocabularies (SaProt (Su et al., 2023)) or to predict structure tokens directly (ISM (Ouyang-Zhang et al., 2025)). These models provide transferable features that support many modern PPI predictors, including our method.

**Protein–Protein Interaction Prediction.** An ideal way to assess PPIs is to predict the 3D structure of protein complexes directly using end-to-end structure predictors such as AlphaFold2 (Jumper et al., 2021), RoseTTAFold (Baek et al., 2021), AlphaFold-Multimer (Evans et al., 2021), and the more recent AlphaFold3 (Jumper et al., 2024). These systems achieve remarkable accuracy but are computationally intensive, often rely on multiple sequence alignments (MSAs) and/or structural templates, and require explicit per-pair inference. Another line of work formulates PPI as a graph machine learning problem (e.g., link prediction (Nasiri et al., 2021)) on residue- or protein-level graphs. Examples include diffusion-state methods (Devkota et al., 2020) and GNN-based approaches such as GNN-PPI (Lv et al., 2021), SGPPI (Huang et al., 2023), PPI-GNN (Jha et al., 2022), and HIGH-PPI (Gao et al., 2023). Although graph priors can be powerful, the performance of such models depends on the availability and quality of the underlying network and can be vulnerable to degree bias and data leakage (Bernett et al., 2024). These limitations prevent proteome-level screening. In light of this, we focus on sequence-only methods, as they are alignment-free (no MSA needed) and graph-free, while maintaining good performance.

Among sequence-only methods, early work such as SPRINT (Li and Ilie, 2017) computes pair-specific similarities with spaced-seed $k$-mer hashing. Subsequent deep learning encoders include DeepFE-PPI (Yao et al., 2019), the fully connected and LSTM models of Richoux et al. (Richoux et al., 2019), and PIPR's Siamese residual RCNN (Chen et al., 2019). Many later sequence models follow the D-SCRIPT (Sledzieski et al., 2021) paradigm, computing residue–residue interaction scores and

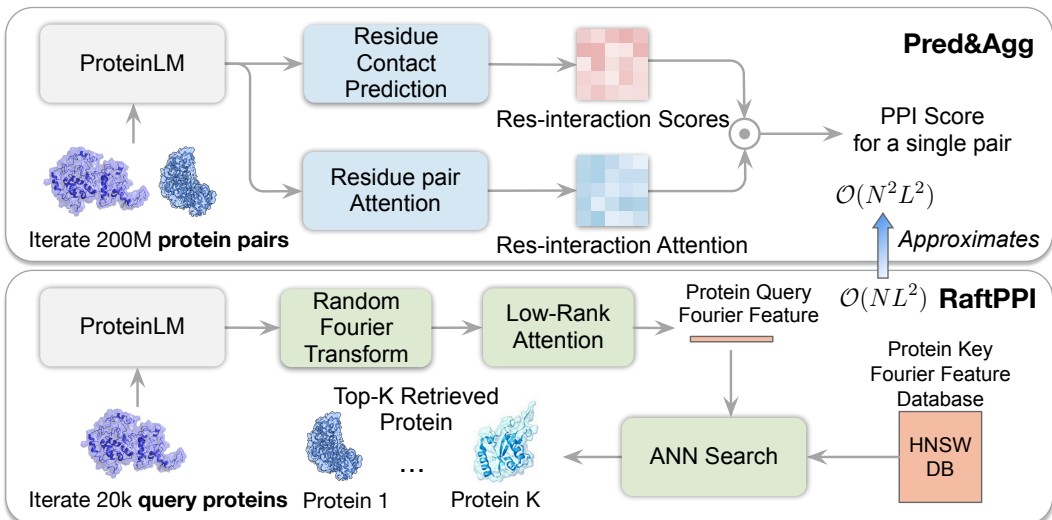

Figure 1: **Overview of RaftPPI.** RaftPPI approximates a standard PPI pipeline *Pred&Agg* (above) that *predicts* residue-level contact scores and *aggregates* them using attention to a final PPI score. The *Pred&Agg* pipeline costs $\mathcal{O}(N^2L^2)$ for proteome-wide retrieval. RaftPPI uses Random Fourier Features to approximate a Gaussian kernel and leverages low-rank attention. The pipeline is factorizable (below), enabling per-protein Fourier-feature embeddings and ultra-fast retrieval via approximate nearest neighbor search (e.g., HNSW) at proteome scale with $\mathcal{O}(NL^2)$ precomputation.

aggregating them into a protein-level score; Topsy-Turvy (Singh et al., 2022) and TT3D (Sledzieski et al., 2023) further extend this approach by incorporating graph priors (Devkota et al., 2020) and structural embeddings (van Kempen et al., 2024). Recently, PLM-based approaches (Sledzieski et al., 2024; Ko et al., 2024; Yang et al., 2024; Liu et al., 2024) leverage the strong performance of pretrained PLMs (Lin et al., 2023; Elnaggar et al., 2021) to model interactions at residue-level resolution. These approaches are typically accurate but *non-factorizable*: they jointly encode each protein pair, which makes proteome-scale retrieval computationally prohibitive.

**Proteome-level PPI Screening.** Whole-proteome screening poses a computational challenge due to the quadratic number of protein pairs. Recent pipelines address this using multi-stage filters coupled with structure prediction. In RF2-style workflows (Humphreys et al., 2024; Zhang et al., 2024), GPU-accelerated coevolutionary analysis (DCA (Ekeberg et al., 2013)) first prunes hundreds of millions of candidates; top pairs are then rescored with a lightweight RoseTTAFold2-based contact predictor, and the highest-confidence subset is finally evaluated with full AlphaFold2. These efforts primarily focus on accelerating structure prediction; for example, RF2-Lite (Humphreys et al., 2024) streamlines refinement with a two-track network that is $\sim 20\times$ faster than AlphaFold2, yet still requires explicit per-pair inference after pre-screening.

**Our Position.** RaftPPI is a residue-aware yet retrieval-friendly framework. It produces indexable protein embeddings whose inner products approximate residue-level interaction scoring, enabling ANN-based proteome screening without explicit per-pair inference. Compared with sequence-only models, RaftPPI attains stronger PPI classification while scaling to whole-proteome retrieval via precomputed embeddings. Compared with structure-prediction methods, it offers orders-of-magnitude faster screening by narrowing candidates first, after which shortlisted pairs can be rescored with accurate complex structure predictors.

## 3 METHODOLOGY

Figure 1 summarizes the RaftPPI framework. We first revisit a standard two-step pipeline for sequence-based PPI prediction (§3.1), then show how RaftPPI approximates residue-level modeling while *enabling scalable protein retrieval* by coupling a Gaussian-kernel interaction with low-rank (separable) attention and Structured Orthogonal Random Features (SORF) (§3.2). We optimize the

model with an *adaptive negative reweighting* objective (§3.4) and enable proteome-scale search via vector indices over precomputed embeddings (§3.3).

Throughout, we consider a candidate protein pair $(A, B)$ with residues indexed $1, \ldots, L_A$ and $1, \ldots, L_B$, respectively. Bold lowercase symbols (e.g., $\boldsymbol{z}$) denote vectors, bold uppercase symbols (e.g., $\boldsymbol{W}$) denote matrices, $\sigma(\cdot)$ is the logistic sigmoid, and $\| \cdot \|$ is the Euclidean norm.

## 3.1 A PIPELINE FOR TWO-STEP PPI PREDICTION

Since protein interactions occur at the residue level, a standard approach is a two-step pipeline (Sledzieski et al., 2021; Singh et al., 2022; Sledzieski et al., 2023) as illustrated in the upper panel of Figure 1: *Predict* residue-pair contact scores, then *aggregate* them into a protein-pair score. We refer to this two-step pipeline as *Pred&Agg* and define it as follows:

**Residue-level contact matrix prediction.** Given residue embeddings $\boldsymbol{z}_{A,i}, \boldsymbol{z}_{B,j} \in \mathbb{R}^d$ from a pretrained protein language model (PLM) (Lin et al., 2023), we compute *contact scores*

$$c_{i,j} = f(\boldsymbol{z}_{A,i}, \boldsymbol{z}_{B,j}) \in \mathbb{R}, \tag{1}$$

where $f(\cdot, \cdot)$ is typically an MLP or an inner product, producing a contact matrix $\boldsymbol{C} \in \mathbb{R}^{L_A \times L_B}$.

**Protein-level interaction aggregation.** This residue-level score matrix is pooled into a scalar logit:

$$\ell(A, B) = g(\boldsymbol{C}) \in \mathbb{R}, \tag{2}$$

where $g(\cdot)$ denotes a pooling operation (e.g., max/conv/2D attention).

There are many instances of this *Pred&Agg* pipeline. For example, D-SCRIPT (Sledzieski et al., 2021) uses Bepler & Berger embeddings (Bepler and Berger, 2019) with element-wise transforms to predict $\boldsymbol{C}$, then pools to a pair score. TT3D (Sledzieski et al., 2023) and Topsy-Turvy (Singh et al., 2022) incorporate graph-based supervision (Devkota et al., 2020) and 3Di structure encodings (van Kempen et al., 2024). While effective, these methods are *non-factorizable*: they rely on dense nonlinear residue–pair computations, yielding $\mathcal{O}(N^2 L^2)$ complexity for $N$ proteins of length $L$, which is prohibitive for proteome-scale screening. Next we show how RaftPPI approximates this pipeline as a dot product between *single-protein* embeddings.

## 3.2 KERNELIZED RESIDUE INTERACTIONS WITH LOW-RANK ATTENTION

As noted above, the nonlinearity in *Pred&Agg* yields strong accuracy but hinders retrieval because it requires explicit *pairwise* inputs. Our idea is to approximate these steps with *factorizable* ones where protein interaction scores are computed via dot-product between protein embeddings.

**Predict (kernelized residue–residue scoring).** We model residue–residue interactions with a Gaussian kernel

$$k_{\hat{\sigma}}(\boldsymbol{z}_{A,i}, \boldsymbol{z}_{B,j}) = \exp\left(-\frac{\|\boldsymbol{z}_{A,i} - \boldsymbol{z}_{B,j}\|^2}{2\hat{\sigma}^2}\right), \tag{3}$$

where $\hat{\sigma}^2$ is the kernel bandwidth that controls how quickly the kernel decays with residue-embedding distance. Smaller $\hat{\sigma}$ emphasizes very local residue matches, while larger values blend information across a wider neighborhood (see Appendix B.2 for the quantitative sweep). The kernel corresponds to an inner product in an infinite-dimensional Reproducing kernel Hilbert space (RKHS), enabling rich nonlinear scoring.

**Aggregate (attention-weighted pooling).** We aggregate the residue-level kernel scores into a protein-level logit via a weighted sum:

$$\ell(A, B) = \sum_{i=1}^{L_A} \sum_{j=1}^{L_B} s_{i,j} k_{\hat{\sigma}}(\boldsymbol{z}_{A,i}, \boldsymbol{z}_{B,j}), \tag{4}$$

where $s_{i,j}$ is the *attention weight*, determining the set of residue pairs of interest.

**Factorizable low-rank attention.**  Low-rank (separable) attention is a standard strategy for reducing quadratic cost (e.g., Wang et al., 2020; Choromanski et al., 2021). We approximate the residue–pair attention with a rank-$r$ separable form. Denote the residue embedding matrix as $\mathbf{Z}_A \in \mathbb{R}^{L_A \times d}$ and $\mathbf{Z}_B \in \mathbb{R}^{L_B \times d}$; for each $t \in \{1, \dots, r\}$ a lightweight per-residue scorer $h_\theta^{(t)} : \mathbb{R}^d \to \mathbb{R}$ is applied *row-wise* to produce unnormalized importances, which we normalize with a softmax to obtain per-chain weights:

$$\mathbf{w}_A^{(t)} = \mathrm{softmax}\big(h_\theta^{(t)}(\mathbf{Z}_A)\big), \qquad \mathbf{w}_B^{(t)} = \mathrm{softmax}\big(h_\theta^{(t)}(\mathbf{Z}_B)\big). \tag{5}$$

Collecting columns gives $\mathbf{W}_A = [\,\mathbf{w}_A^{(1)} \cdots \mathbf{w}_A^{(r)}\,] \in \mathbb{R}_{\geq 0}^{L_A \times r}$ and $\mathbf{W}_B = [\,\mathbf{w}_B^{(1)} \cdots \mathbf{w}_B^{(r)}\,] \in \mathbb{R}_{\geq 0}^{L_B \times r}$, with nonnegative entries and each column summing to $1$. We then set

$$s_{i,j} = \sum_{t=1}^{r} w_{A,i}^{(t)} w_{B,j}^{(t)} \quad \Longleftrightarrow \quad \mathbf{S} = \mathbf{W}_A \mathbf{W}_B^\top, \tag{6}$$

yielding a *factorizable* attention surface. In practice we instantiate $r=1$ (so $s_{i,j} = w_{A,i} w_{B,j}$), which achieves strong performance with minimal parameter cost.

## 3.3   Fast Inference with Random Fourier Features and Vector Search

**Kernel approximation with Random Fourier Features.**  We approximate $k_{\hat{\sigma}}$ using Random Fourier Features (RFF) (Rahimi and Recht, 2007). Given $d'$ target frequencies, let $\mathbf{W} \in \mathbb{R}^{d' \times d}$ and define

$$\psi(\mathbf{z}) = \tfrac{1}{\sqrt{d'}} \big[\cos(\mathbf{W}\mathbf{z}); \sin(\mathbf{W}\mathbf{z})\big] \in \mathbb{R}^{2d'}, \tag{7}$$

so that we have a factorized form to approximate the kernel as:

$$k_{\hat{\sigma}}(\mathbf{x}, \mathbf{y}) \approx \psi(\mathbf{x})^\top \psi(\mathbf{y}). \tag{8}$$

To construct $\mathbf{W}$ efficiently, we use Structured Orthogonal Random Features (SORF) (Yu et al., 2016). A SORF block of size $d \times d$ is

$$\tilde{\mathbf{W}} = \tfrac{\sqrt{d}}{\hat{\sigma}} \, \mathbf{H}\mathbf{D}_1\mathbf{H}\mathbf{D}_2\mathbf{H}\mathbf{D}_3, \tag{9}$$

where $\mathbf{H}$ is the normalized Walsh–Hadamard matrix and $\mathbf{D}_i$ are diagonal Rademacher sign-flip matrices. The rows of $\tilde{\mathbf{W}}$ satisfy $\mathbb{E}[\mathbf{w}\mathbf{w}^\top] = \hat{\sigma}^{-2}\mathbf{I}$, matching the Gaussian second moment and providing a low-variance RFF approximation of the Gaussian kernel. To obtain $d'$ frequencies, we generate independent SORF blocks and concatenate their first $d'$ rows to form $\mathbf{W} \in \mathbb{R}^{d' \times d}$.

**Fast retrieval via factorizable scoring.**  Using Eq. 8 and the $r=1$ attention, we obtain

$$\begin{aligned}
\ell(A, B) &= \sum_{i=1}^{L_A} \sum_{j=1}^{L_B} s_{i,j}\, k_{\hat{\sigma}}(\mathbf{z}_{A,i}, \mathbf{z}_{B,j}) \\
&\approx \sum_{i=1}^{L_A} \sum_{j=1}^{L_B} w_{A,i} w_{B,j}\, \psi(\mathbf{z}_{A,i})^\top \psi(\mathbf{z}_{B,j}) \\
&= \Big\langle \sum_{i=1}^{L_A} w_{A,i}\, \psi(\mathbf{z}_{A,i}), \sum_{j=1}^{L_B} w_{B,j}\, \psi(\mathbf{z}_{B,j}) \Big\rangle.
\end{aligned} \tag{10}$$

Define the per-protein embeddings as

$$\hat{\mathbf{h}}_A = \sum_{i=1}^{L_A} w_{A,i}\, \psi(\mathbf{z}_{A,i}), \qquad \hat{\mathbf{h}}_B = \sum_{j=1}^{L_B} w_{B,j}\, \psi(\mathbf{z}_{B,j}), \tag{11}$$

which yields the factorizable approximation that computes the logit $\ell(A, B)$ in a dot-product form:

$$\ell(A, B) := \langle \hat{\mathbf{h}}_A, \hat{\mathbf{h}}_B \rangle. \tag{12}$$

We fix the same SORF transform $\mathbf{W}$ at training and inference for alignment, and store $\hat{\mathbf{h}}$ for approximate $k$-nearest neighbor search with HNSW (Malkov and Yashunin, 2020) (inner-product retrieval in the transformed space).

### 3.4 TRAINING OBJECTIVE

As noted in the Introduction, experimentally verified non-interactions are rare, so negatives are often *constructed* and vary widely in informativeness (Neumann et al., 2022; Zhao et al., 2022). Heuristic constructions (e.g., enforcing different cellular compartments) often produce overly easy, biased negatives, leading to overly optimistic results. Meanwhile, co-localized, functionally related, or topology-aware choices tend to be harder and more informative (Ben-Hur and Noble, 2006; Park and Marcotte, 2011; 2012; Zhang et al., 2018). Motivated by this, we adopt *adaptive negative weighting*, where the relative contribution of each negative is automatically determined by the model's own confidence, allowing harder negatives to exert greater influence.

Let $\ell(A, B) = \langle \hat{\boldsymbol{h}}_A, \hat{\boldsymbol{h}}_B \rangle$ denote the logit for a pair $(A, B)$. Over a minibatch, let $\mathcal{P}$ and $\mathcal{N}$ be the index sets of positive and negative pairs, respectively. Inspired by self-adversarial training in knowledge-graph reasoning (Sun et al., 2019), we define temperature-scaled weights over negatives

$$p_i = \frac{\exp\left(\tau\, \ell_i\right)}{\sum\limits_{j \in \mathcal{N}} \exp\left(\tau\, \ell_j\right)}, \qquad i \in \mathcal{N}, \quad \tau \geq 0, \tag{13}$$

and stop gradients through $p_i$ in practice. Intuitively, $p_i$ reflects the model's (normalized) confidence that a negative pair is actually positive, i.e. higher $p_i$ thus identifies *harder* negatives. We then combine a standard positive term with an adaptively reweighted negative term:

$$\mathcal{L} = \frac{1}{2} \left[ -\frac{1}{|\mathcal{P}|} \sum_{p \in \mathcal{P}} \log \sigma(\ell_p) - \sum_{i \in \mathcal{N}} p_i \, \log \sigma(-\ell_i) \right]. \tag{14}$$

When $\tau = 0$, Eq. 14 reduces to balanced BCE; as $\tau \to \infty$, it focuses on the hardest negative. In practice, $\tau = 4$ offers a good trade-off, as shown in Appendix B.4.

### 3.5 COMPUTATIONAL COMPLEXITY

Consider a proteome with $N$ proteins of average length $L$. PLM embedding dominates at $\mathcal{O}(NL^2)$. Our mapping/aggregation adds $\mathcal{O}\left(L\left(d \log d + d'\right)\right)$ per protein (vs. $\mathcal{O}(Ldd')$ for dense RFF), which is linear in $L$ and minor in practice. After caching $\hat{\boldsymbol{h}}$, HNSW indexing is $\mathcal{O}(N \log N)$, queries grow polylogarithmically in $N$, and memory is $\mathcal{O}(N)$. Hence the end-to-end complexity is $\mathcal{O}(NL^2)$, with indexing/search negligible; empirically, top-20% retrieval completes in $\sim$6 minutes on an A100 (, or 3.3 minutes on an Intel Xeon 6 CPU) for $N \approx 10^4$ (see §4.3, Table 3).

## 4 EXPERIMENTS

### 4.1 EXPERIMENTAL SETUP

**Datasets.** As discussed in (Bernett et al., 2024), naïve PPI splits and datasets are prone to *data leakage* and confounding from sequence similarity and node-degree biases, which can yield over-optimistic performance and poor transfer. We therefore adopt the 7 processed datasets/splits in (Bernett et al., 2024) that (i) remove near-duplicate or homologous sequences across train/validation and test sets, minimizing the impact of raw sequence similarity, and (ii) control protein occurrence frequency so that hub proteins do not trivially inflate accuracy via degree priors. This setting makes PPI prediction more realistic and challenging. The datasets span two species: yeast (Guo, Du) and human (Huang, D-SCRIPT, Pan, Richoux, Gold). Their statistics are shown in Table 1. In Appendix B.6 we further evaluate RaftPPI and baselines on the larger-scale PiNUI-human and PiNUI-yeast datasets (Dubourg-Felonneau et al., 2023).

**Baselines.** We evaluate 10 PPI classifiers spanning classical sequence models (D-SCRIPT (Sledzieski et al., 2021), DeepFE (Yao et al., 2019), Richoux-FC/LSTM (Richoux et al., 2019), Topsy–Turvy (Singh et al., 2022), SPRINT (Li and Ilie, 2017)) and PLM-based methods (ESM2-MLP (Sledzieski et al., 2024), TUnA (Ko et al., 2024), PLM-Interact (Liu et al., 2024)). For all PLM baselines and for RaftPPI, we use ESM2-8M as the backbone; we also include an unsupervised ESM2-NoFT baseline (dot product over `[CLS]` embeddings). Prior work (Fournier et al., 2024)

Table 1: Dataset statistics of seven PPI datasets spanning two species (human and yeast).

| Dataset | Species | Train | | | Val | | | Test | | | Total |
|---|---|---|---|---|---|---|---|---|---|---|---|
| | | Pos | Neg | Total | Pos | Neg | Total | Pos | Neg | Total | |
| GUO | Yeast | 2,088 | 2,088 | 4,176 | 232 | 232 | 464 | 861 | 861 | 1,722 | 6,362 |
| DU | Yeast | 6,536 | 6,486 | 13,022 | 698 | 748 | 1,446 | 2,421 | 2,421 | 4,842 | 19,310 |
| HUANG | Human | 1,094 | 1,075 | 2,169 | 111 | 130 | 241 | 713 | 713 | 1,426 | 3,836 |
| D-SCRIPT | Human | 12,218 | 122,165 | 134,383 | 1,356 | 13,575 | 14,931 | 8,467 | 84,670 | 93,137 | 242,451 |
| PAN | Human | 14,069 | 14,022 | 28,091 | 1,537 | 1,584 | 3,121 | 4,575 | 4,575 | 9,150 | 40,362 |
| RICHOUX | Human | 17,873 | 17,798 | 35,671 | 1,944 | 2,019 | 3,963 | 5,167 | 5,167 | 10,334 | 49,968 |
| GOLD | Human | 81,596 | 81,596 | 163,192 | 29,630 | 29,630 | 59,260 | 26,024 | 26,024 | 52,048 | 274,500 |

and our analysis at Appendix B.1 show that scaling ESM2 (35M/150M/650M) does not consistently improve these tasks; moreover, proteome-scale retrieval with per-pair PLM inference is already expensive at the 8M scale, consuming GPU-months computation time (see Table 3). In this case, we use ESM2-8M for the best performance/throughput trade-off in our evaluations.

## 4.2 PROTEIN-PROTEIN INTERACTION CLASSIFICATION

Table 2: Test AUROC Performance (%) of competing methods on the seven PPI datasets. Higher is better; the rightmost column shows the mean across collections.

| Method | D-SCRIPT | Huang | Pan | Richoux | Gold | Guo | Du | Average |
|---|---|---|---|---|---|---|---|---|
| D-SCRIPT (Sledzieski et al., 2021) | 81.99 | 65.72 | 68.44 | 59.15 | 49.91 | 47.14 | 50.92 | 60.47 |
| DeepFE (Yao et al., 2019) | 61.66 | 56.93 | 53.70 | 57.43 | 53.21 | 58.21 | 56.05 | 56.74 |
| Richoux-FC (Richoux et al., 2019) | 47.53 | 59.06 | 52.39 | 59.72 | 53.53 | 55.81 | 59.89 | 55.42 |
| Richoux-LSTM (Richoux et al., 2019) | 50.67 | 56.56 | 47.81 | 50.37 | 49.16 | 51.55 | 56.37 | 51.78 |
| SPRINT (Li and Ilie, 2017) | 64.62 | 46.71 | 43.39 | 55.71 | 51.50 | 48.80 | 51.80 | 51.79 |
| Topsy-Turvy (Singh et al., 2022) | 75.38 | 55.52 | 65.80 | 48.67 | 58.74 | 43.65 | 61.60 | 58.48 |
| ESM2-NoFT (Lin et al., 2023) | 75.01 | 58.63 | 59.51 | 63.26 | 57.85 | 62.87 | 57.36 | 62.07 |
| ESM2-MLP (Sledzieski et al., 2024) | 82.83 | **73.34** | 72.89 | 77.48 | 56.35 | 83.54 | 73.34 | 74.25 |
| TUnA (Ko et al., 2024) | 83.38 | 66.66 | **77.06** | 76.73 | 52.55 | 69.81 | 69.37 | 70.79 |
| PLM-Interact (Liu et al., 2024) | **84.77** | 69.69 | 73.03 | **78.66** | 65.00 | 79.60 | **75.20** | 75.14 |
| RaftPPI | 82.06 | 72.20 | 74.21 | 69.88 | **68.69** | **84.93** | 75.06 | **75.29** |

The PPI classification results across the seven datasets are reported in Table 2. We observe that methods without pretrained PLMs—D-SCRIPT, DeepFE, Richoux-FC/LSTM, Topsy-Turvy, and SPRINT—perform substantially worse than ESM2-based models, all falling below even the unsupervised ESM2-NoFT baseline. This is because they rely largely on sequence similarity and node-degree information, which fails under controlled splits (Bernett et al., 2024), whereas ESM-based models benefit from large-scale pretraining, where structural properties such as secondary structure can be inferred from embeddings (Rives et al., 2021).

For ESM2-based baselines, all models finetuned for PPI outperform ESM2-NoFT. PLM-Interact achieves the strongest results, likely due to its early-fusion design, which allows deeper layers to jointly model cross-protein interactions. In contrast, TUnA and ESM2-MLP fuse only at intermediate or final layers, limiting their ability to capture joint interactions. This echoes the early-fusion advantage reported in other domains (Snoek et al., 2005). Meanwhile, RaftPPI attains the best average performance, attributable to residue-level interaction modeling and adaptive negative weighting, which we further discuss in §4.4.

## 4.3 PROTEOME INTERACTION RETRIEVAL

Compared with binary PPI classification, PPI retrieval more closely reflects real-world applications: interactions in proteomes are *sparse* and highly *imbalanced* (negatives dominate), and one must screen an entire candidate proteome to identify true interactors for a query protein. For computational tractability, we sample 100 query proteins per dataset and use the models trained in Section 4.2 to retrieve positives on the test split. We compare RaftPPI to PLM (ESM2) baselines (ESM2-NoFT, TUnA, PLM-Interact, ESM2-MLP) and to RaftPPI-P, a special version that removes the residue-level design of RaftPPI, which predicts PPI using the dot-product of `[CLS]` token embeddings.

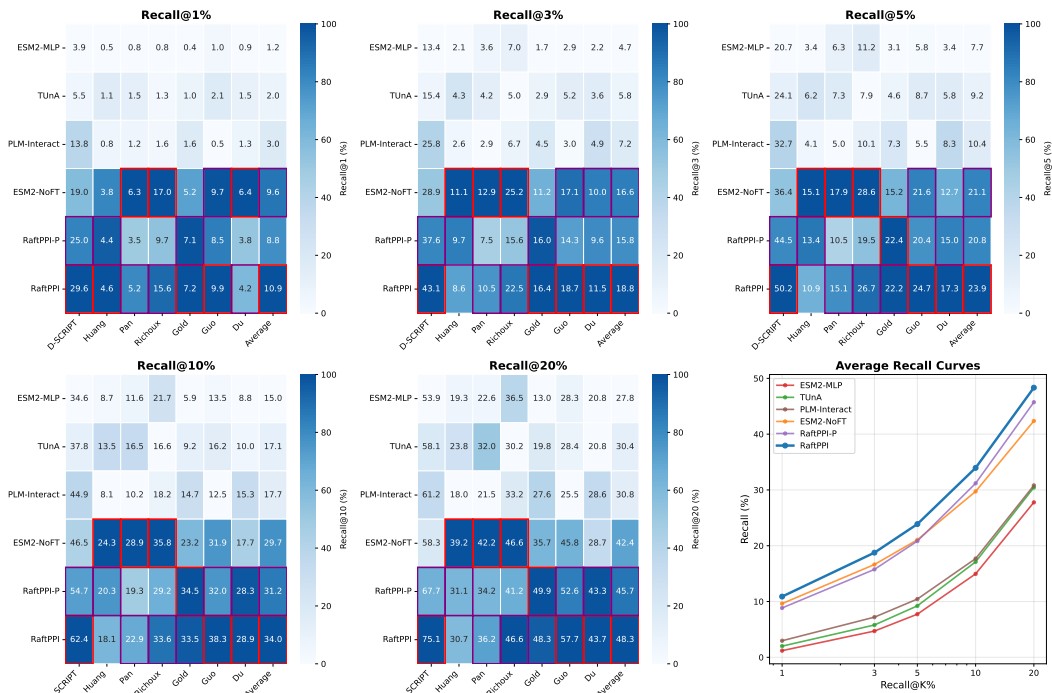

Figure 2: **Proteome retrieval with residue-level fidelity and scalable retrieval.** Heatmaps report Recall@$K$% for $K \in \{1, 3, 5, 10, 20\}$ across methods and datasets, with per-dataset *normalized* color scales (best=100%, worst=0%). Red rectangles mark the best method; purple rectangles mark the second best. The sixth panel shows average recall curves across datasets.

| Type | Model | Encoding (s) | Recall@1/3/5/10/20% (s) | AUROC (%) | Recall@20% est. / full (%) |
|---|---|---|---|---|---|
| Unfactorizable | ESM2-MLP | NA | 1766576 | 74.25 | 27.77 / NA |
| | TUnA | NA | 3833646 | 70.79 | 30.44 / NA |
| | PLM-Interact | NA | 12827660 | 75.14 | 30.81 / NA |
| Factorizable | ESM2-NoFT | 105 | 54 / 74 / 99 / 170 / 259 | 62.07 | 42.37 / 41.72 |
| | RaftPPI-P | 54 | 40 / 48 / 75 / 118 / 187 | 71.90 | 45.73 / 43.83 |
| | RaftPPI | 102 | 49 / 70 / 98 / 157 / 241 | **75.29** | **48.33 / 47.91** |

Table 3: Human proteome retrieval efficiency (Recall@K% end-to-end time) and average classification/retrieval performance. Factorizable methods reuse single-protein embeddings to build the HNSW index once (encoding time is a one-time cost). For unfactorizable methods, recall (est.) and time are estimated using 100 query proteins; reported times are total seconds for the full proteome, measured on one A100 GPU. More runtime settings and discussion are in Appendix A.

**Retrieval performance.** As shown in Figure 2, non-factorizable methods—i.e., models that jointly encode protein pairs such as ESM2-MLP, TUnA, and PLM-Interact—achieve strong binary PPI classification but do not outperform the simple ESM2-NoFT baseline in retrieval. We hypothesize that this gap arises because the data splits are explicitly designed to minimize sequence similarity (Bernett et al., 2024), limiting the models' ability to exploit correlations between sequence similarity and structural interaction. In contrast, factorizable approaches naturally capture sequence similarity through embedding dot products. Among these, RaftPPI consistently ranks first or second across datasets and recall thresholds, and its improvements over RaftPPI-P highlight the benefit of explicitly modeling residue-level interactions.

**Retrieval efficiency.** Table 3 reports runtime comparisons. Non-factorizable methods—ESM2-MLP, TUnA, and PLM-Interact—are prohibitively slow because they require per-pair inference. The strongest of these, PLM-Interact, performs well on both classification and retrieval, yet demands 148.47 A100 GPU-days ($\approx$4.9 months) to screen the human proteome, underscoring the impracticality of exhaustive search. By contrast, RaftPPI compresses each protein once and performs approximate

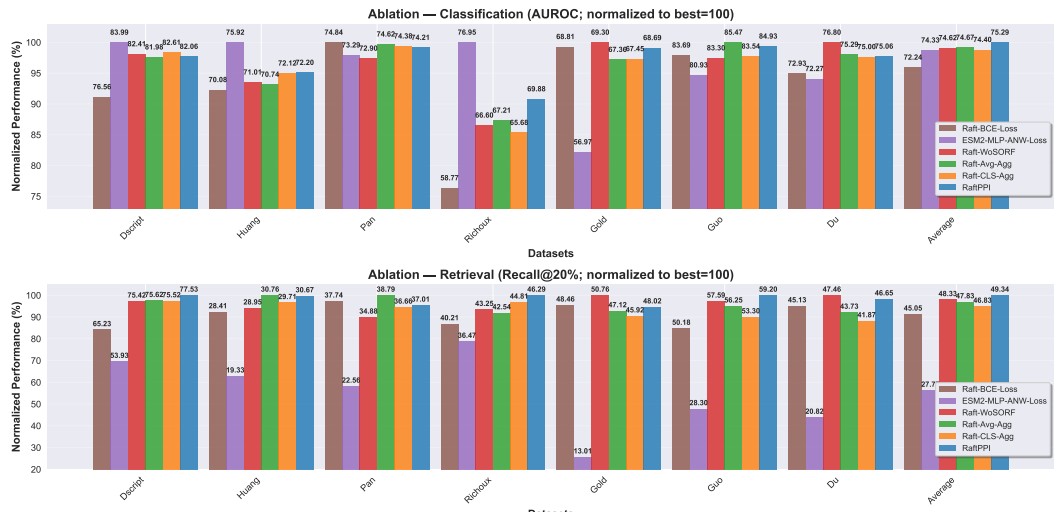

Figure 3: **Ablation Study.** Scores are normalized to the best method per dataset (100%=peak). *Top:* classification AUROC. *Bottom:* retrieval Recall@20%. The full model (RaftPPI) attains the best or second–best performance in every case; removing any single design choice causes a clear drop.

nearest-neighbor search, reducing proteome retrieval from *GPU months* to *minutes*. Overall, RaftPPI delivers the best balance of retrieval accuracy, classification performance, and efficiency.

### 4.4 ABLATION STUDY

We focus on four research questions and validate our core designs. **RQ1 (residue-level modeling):** Does explicit residue-level modeling help? **RQ2 (kernel):** Does replacing a linear dot product with a Gaussian kernel (approximated via SORF) improve performance? **RQ3 (aggregation):** How should residue scores be aggregated most effectively? **RQ4 (adaptive negative weighting):** Is adaptive negative weighting effective in helping the model assign greater weight to harder negatives?

We perform ablation studies on these research questions and show the results in Figure 3. **RQ1 (residue-level modeling):** The coarse ESM2-MLP baseline encodes proteins and predicts interactions from concatenated embeddings. Even when augmented with our adaptive loss (ESM2-MLP-ANW-Loss), it competes on small classification splits but collapses on retrieval (e.g., $-46\%$ Recall@20% on Gold), confirming the necessity of residue-level reasoning for large-candidate screening. **RQ2 (kernel):** Replacing the Gaussian kernel with a linear dot product (Raft-WoSORF) consistently reduces AUROC and Recall@20%, showing the benefit of kernelized interactions. We additionally discuss the impact of Gaussian bandwidth $\hat{\sigma}$ in Appendix B.2. **RQ3 (aggregation):** Averaging (Raft-Avg-Agg) or using a `[CLS]` score (Raft-CLS-Agg) are slightly weaker than attention, indicating that attention helps identify PPIs. **RQ4 (adaptive negative weighting):** Switching to uniform BCE (Raft-BCE) degrades AUROC and Recall@20%, especially on the D-SCRIPT dataset where negatives outnumber positives by roughly $10\times$, highlighting the value of prioritizing hard negatives (we provide a detailed ablation study of the temperature $\tau$ in Appendix B.4). Collectively, these ablations validate each design choice in RaftPPI, demonstrating the effectiveness of kernelized residue interactions, SORF-based kernels, attention-based aggregation, and adaptive negative weighting.

## 5 CONCLUSION

We introduced RaftPPI, a residue-aware yet retrieval-friendly framework that factorizes kernelized residue interactions with structured random Fourier features and low-rank attention to produce compact per-protein embeddings for ANN search. With adaptive negative weighting, RaftPPI achieves the best average classification and retrieval performance across seven sequence- and degree-controlled benchmarks while enabling minutes-level screening on the human proteome.

ACKNOWLEDGMENT

This project is supported by the Intel-MILA partnership program, the Natural Sciences and Engineering Research Council (NSERC) Discovery Grant, the Canada CIFAR AI Chair Program, collaboration grants between Microsoft Research and Mila, and a NRC Collaborative R&D Project (AI4DCORE-06). This project was also partially funded by IVADO Fundamental Research Project grant PRF-2019-3583139727. The computation resource of this project is supported by Mila, Calcul Québec and the Digital Research Alliance of Canada and Intel Corporation. For several intermediate approaches, a cluster consisting of Intel Xeon Scalable CPUs (Intel® Xeon® CPU Max 9470C) and Intel Max GPUs (Intel® Data Center Max 1550) was used for training and inference.

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

## A    IMPLEMENTATION DETAILS

**Experimental settings.** All experiments use five random seeds $\{0, 1, 2, 3, 4\}$; unless noted, we report the mean (and standard deviation when available) across seeds. Results for D-SCRIPT (Sledzieski et al., 2021), DeepFE (Yao et al., 2019), Richoux-FC/LSTM (Richoux et al., 2019), and Topsy-Turvy (Singh et al., 2022) are taken from the benchmarking study of Bernett et al. (2024); following their guidance, we apply minimal tuning to baselines based on validation performance. Unless otherwise specified, we use a single configuration across datasets: AdamW (lr $10^{-4}$), 2048 random Fourier features, Gaussian kernel bandwidth $\hat{\sigma} = 0.5$ (App. B.2), and adversarial temperature $\tau = 4$ (App. B.4). Software environment: Python 3.10; PyTorch 2.5 with CUDA 11.8.

**Hardware and runtime reporting.** GPU timings in Table 3 are measured on a single NVIDIA A100 GPU. CPU timings are measured on an Intel® Xeon® 6980P consisting of 128 cores.[1] In the human-proteome experiment in §4.3 ($N \approx 20{,}000$ proteins), retrieving top-20% candidates corresponds to querying $K = 0.2N$ nearest neighbors per protein (about 4,000), yielding $\approx 40$M unique pairs after symmetrization. While non-factorizable methods must iterate over all candidate pairs ($\approx 200$M), RaftPPI enables fast proteome-scale retrieval via a pre-built HNSW index over cached single-protein embeddings. On CPU, the end-to-end time is 200 seconds (3.3 minutes): 71 seconds for encoding all proteins and 129 seconds for top-20% HNSW retrieval. On GPU, the end-to-end time is 343 seconds (5.7 minutes): 102 seconds encoding and 241 seconds retrieval (Table 3).

## B    ADDITIONAL EXPERIMENTS

### B.1    MODEL-SCALE SELECTION

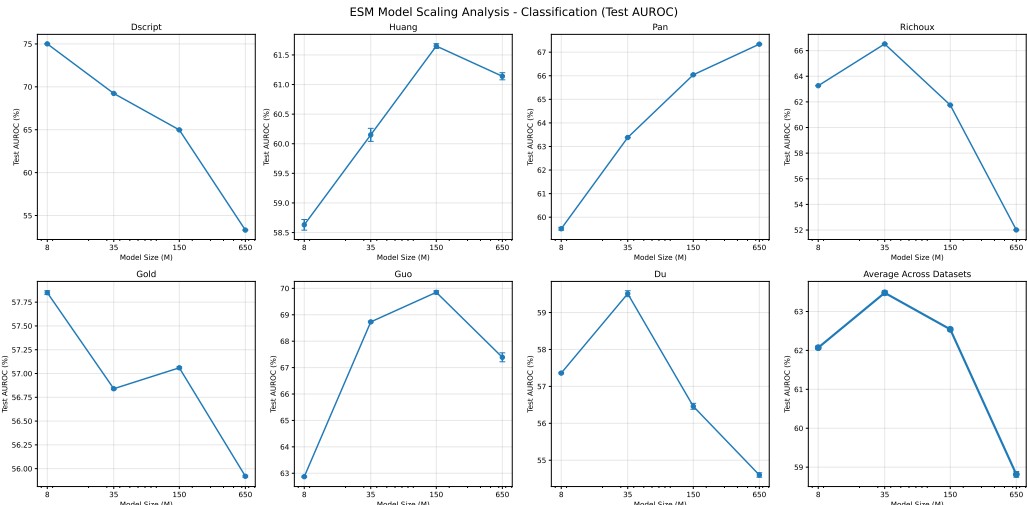

Figure 4: ESM2 model–scaling analysis for PPI binary-classification (Test AUROC) across model sizes and datasets. Increasing the parameter count beyond 8M does not consistently improve performance.

Following the observation of Fournier et al. (2024) that larger protein LMs do not necessarily yield better results, we conduct a systematic scaling study on ESM2 (Lin et al., 2023). We evaluate four

---

[1] https://www.intel.com/content/www/us/en/products/sku/240777/intel-xeon-6980p-processor-504m-cache-2-00-ghz/specifications.html

Optimization Notice: Software and workloads used in performance tests may have been optimized for performance only on Intel microprocessors. Performance tests, such as SYSmark and MobileMark, are measured using specific computer systems, components, software, operations and functions. Any change to any of those factors may cause the results to vary. You should consult other information and performance tests to assist you in fully evaluating your contemplated purchases, including the performance of that product when combined with other products. For more information go to http://www.intel.com/performance. Intel, Xeon, and Intel Xeon Phi are trademarks of Intel Corporation in the U.S. and/or other countries.

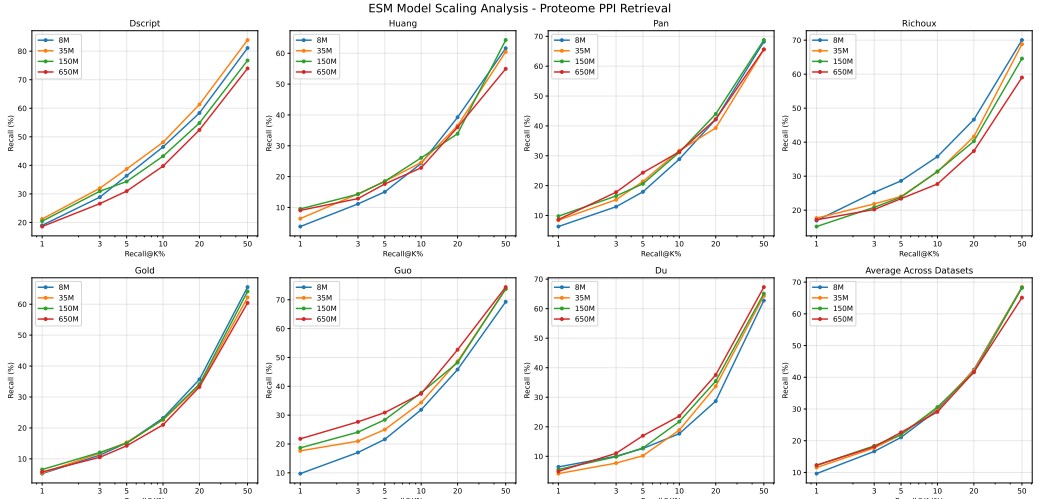

Figure 5: Scaling analysis of ESM2 checkpoints on proteome-level PPI retrieval tasks (Recall@K%). Performance remains largely unchanged when increasing model size.

checkpoints: 8M, 35M, 150M, and 650M parameters across seven PPI datasets spanning human and yeast. Protein pairs are scored by the dot product of their `[CLS]` embeddings, providing an unsupervised measure of scaling performance. As shown in Figures 4 and 5, both Test AUROC and Recall@K% change little with model size on nearly all datasets.

Given that inference time for pairwise-encoding models rises steeply with both model size and quadratic pairwise scoring (e.g., PLM-Interact (Liu et al., 2024) requires *GPU-months* to search the human proteome even with an 8M model due to its pairwise encoding; see Table 3 ), we adopt the 8M ESM2 checkpoint as the backbone for all PLM-based baselines and for RaftPPI, balancing predictive performance and efficiency.

## B.2  ABLATION ON GAUSSIAN KERNEL BANDWIDTH

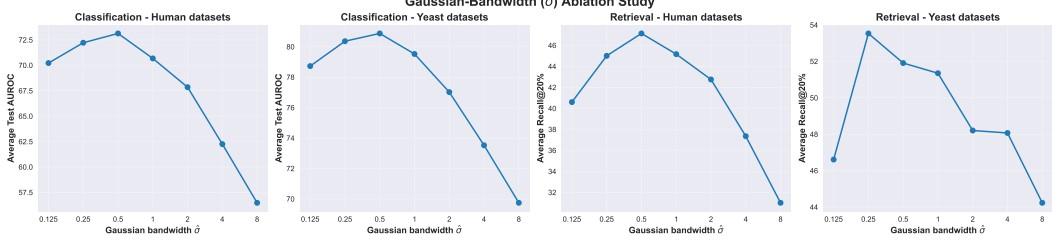

Figure 6: Gaussian-bandwidth ablation: We vary the kernel width $\hat{\sigma}$ from $0.125$ to $8$ (log-scale $x$-axis) and report the averaged AUROC/Recall@20% across human and yeast datasets.

The Gaussian kernel in Eq. 3 controls how strictly residue-level interactions are determined: smaller $\hat{\sigma}$ values confine each residue to interact only with very close neighbors (capturing sharp, local interfaces), whereas larger $\hat{\sigma}$ allows larger interaction scores over a broader neighborhood. Figure 6 sweeps $\hat{\sigma} \in \{0.125, 0.25, 0.5, 1, 2, 4, 8\}$ and averages the metrics within human and yeast datasets. From the results of both PPI classification and retrieval tasks, both very small and very large bandwidths degrade performance. For example, tiny $\hat{\sigma}$ encourages sparse interactions, yet leads to a clear drop in performance when the bandwidth is too small; very large $\hat{\sigma}$ over-smooths the residue-level structure and significantly hurts both PPI classification and retrieval performance. As $\hat{\sigma} = 0.5$ consistently achieves good performance across species, we adopt this value for all reported experiments.

Table 4: Impact of attention rank on AUROC and retrieval (macro-average across seven benchmarks). Higher is better; bold marks the best value and underline the second best per metric.

| Rank | AUROC (%) | Recall@1% | Recall@3% | Recall@5% | Recall@10% | Recall@20% |
|------|-----------|-----------|-----------|-----------|------------|------------|
| 1 | **75.29** | 10.89 | 18.19 | 23.31 | **33.95** | **48.33** |
| 2 | 74.56 | **11.33** | **18.89** | **24.28** | 33.72 | 47.24 |
| 4 | 69.49 | 10.23 | 17.12 | 22.10 | 31.47 | 43.89 |
| 8 | 66.13 | 8.00 | 13.95 | 18.65 | 27.84 | 40.39 |
| 16 | 63.86 | 7.08 | 12.86 | 17.16 | 25.50 | 37.91 |
| 32 | 64.71 | 7.46 | 12.98 | 17.37 | 25.76 | 38.19 |

### B.3 ABLATION ON LOW-RANK ATTENTION

Section 3.2 defines rank-$r$ attention as $r$ independent softmax pools (columns sum to one), and Eq. 4 combines the pooled descriptors with the Gaussian feature map in Eq. 3 . For completeness, we derive the rank-$r$ retrieval formulation implied by that construction. Let $\boldsymbol{\Psi}_A \in \mathbb{R}^{L_A \times 2d'}$ and $\boldsymbol{\Psi}_B \in \mathbb{R}^{L_B \times 2d'}$ stack the Random Fourier Features for proteins $A$ and $B$, and let the attention matrices be $\boldsymbol{W}_A \in \mathbb{R}^{L_A \times r}$ and $\boldsymbol{W}_B \in \mathbb{R}^{L_B \times r}$ with columns that sum to one. The pooled embeddings for each rank are the rows of

$$\boldsymbol{H}_A = \boldsymbol{W}_A^\top \boldsymbol{\Psi}_A \in \mathbb{R}^{r \times 2d'}, \qquad \boldsymbol{H}_B = \boldsymbol{W}_B^\top \boldsymbol{\Psi}_B \in \mathbb{R}^{r \times 2d'}, \tag{15}$$

and the proteome-scale logit becomes

$$\ell(A, B) = \sum_{t=1}^{r} \langle \boldsymbol{h}_A^{(t)}, \boldsymbol{h}_B^{(t)} \rangle = \mathrm{tr}\big(\boldsymbol{H}_A \boldsymbol{H}_B^\top\big) = \langle \mathrm{vec}(\boldsymbol{H}_A), \mathrm{vec}(\boldsymbol{H}_B) \rangle, \tag{16}$$

where $\mathrm{vec}(\cdot)$ denotes vectorization. Computing retrieval scores for $r > 1$ therefore amounts to concatenating the $r$ heads into a single embedding of dimension $2d'r$ per protein. Higher ranks can provide stronger expressiveness, but the embedding dimension, memory footprint, and dot-product cost all grow linearly with $r$ because each additional head contributes another $2d'$ Random Fourier Features ($d'$ for each sin/cos feature).

Table 4 reports the seven-dataset macro-average. Rank 1 already achieves the best AUROC (75.29) and Recall@20% (48.33). Moving to rank 2 mildly improves the very top of the ranking—Recall@1%, Recall@3%, and Recall@5% increase by less than 1% without enhancing AUROC, indicating that the extra head enables slightly stronger early recall capabilities. Larger ranks, however, consistently overfit: AUROC drops below 70 at rank 4 and to 63.86 at rank 16, while Recall@20% degrades from 48.33 (rank 1) to 37.91 (rank 16) despite the $16\times$ increase in memory footprint. Rank 32 further underperforms on every metric. Since rank-one attention is effective enough while achieving the best efficiency compared to higher ranks, we keep $r = 1$ as the default.

### B.4 ABLATION ON ADAPTIVE NEGATIVE WEIGHTING LOSS

As introduced in § 3.4 , we adopt adaptive negative weighting to mitigate the challenge of constructed negatives in PPI datasets. We use a weighted BCE loss that softly prioritizes harder negatives through a temperature parameter (Eq. 13 ). When $\tau \to 0$, the objective reduces to uniform BCE; when $\tau \to \infty$ , it focuses entirely on the single hardest negative in the batch. Figure 7 shows that neither extreme is optimal. Across species and protocols, performance peaks around $\tau = 4$: increasing $\tau$ from 0.125 to 4 improves average test AUROC by $\approx$ 2–3 points and Recall@20% by $\approx$ 3 points, while larger values degrade results by overfitting to outliers. This ablation highlights that moderately emphasizing harder negatives can improve proteome-scale retrieval.

### B.5 ABLATION ON RANDOM FOURIER FEATURES DIMENSION

The Random Fourier Features (RFF) embedding dimension $d'$ (where $d'$ is the number of frequencies in Eq. 8) controls the expressiveness of the Gaussian kernel approximation. Higher dimensions provide more accurate kernel approximation but increase computational cost and memory footprint linearly with the embedding dimension (the sin/cos feature dimension scales as $\mathcal{O}(2d')$ and storage

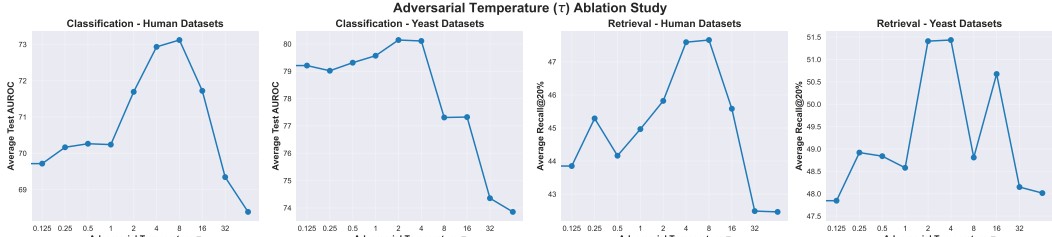

Figure 7: **Adversarial–temperature ablation.** We vary the temperature $\tau$ in the adaptive negative weighting loss (Eq. 14) from $0.125$ to $32$ (log-scale $x$-axis). Each panel reports the average metric over the indicated datasets. A moderate value of $\tau = 4$ (vertical peak) consistently maximizes both classification AUROC (left two panels) and retrieval Recall@20% (right two panels) on human and yeast benchmarks.

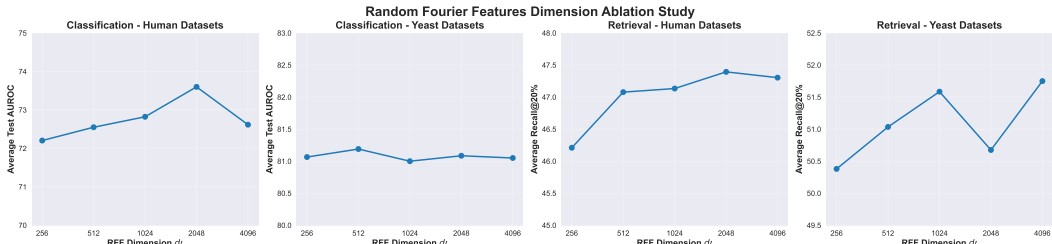

Figure 8: **Random Fourier Features dimension ablation.** We vary the RFF embedding dimension $d'$ from $256$ to $4096$ (log-scale $x$-axis; the corresponding sin/cos feature dimension is $2d'$) and report the averaged AUROC/Recall@20% across human and yeast datasets. In contrast to other hyperparameters, performance remains remarkably stable across all tested dimensions, with changes of less than $1.1$ points, demonstrating that $d'$ is not an important parameter.

scales as $\mathcal{O}(Nd')$ for $N$ proteins). Figure 8 sweeps $d' \in \{256, 512, 1024, 2048, 4096\}$ and averages the metrics within human and yeast datasets.

Compared to the previous hyperparameter ablations, the RFF dimension $d'$ is *not a sensitive parameter* within our tested range. Performance remains remarkably stable across all tested dimensions: varying $d'$ from $256$ to $4096$ changes average AUROC by less than $1.1$ points and Recall@20% by less than $1.1$ points on human datasets, with similarly small variations on yeast datasets. These variations are substantially smaller than the change in performance when varying $\hat{\sigma}$ (Appendix B.2) and varying $\tau$ (Appendix B.4): for example, on human datasets, $\hat{\sigma} = 0.5$ achieves $73.1$ AUROC while $\hat{\sigma} = 8.0$ drops to $56.5$ AUROC, a $16.7$ point difference. The stability of RFF dimension across a $16\times$ range demonstrates that RaftPPI is robust to the RFF dimension choice, and any reasonable value (e.g., $512$–$2048$) works well in practice.

### B.6 EVALUATION ON PiNUI DATASETS

We additionally evaluate our method on the PiNUI datasets (Dubourg-Felonneau et al., 2023) , which provide additional large-scale PPI classification benchmarks for human and yeast. After cleaning, PiNUI-human contains 684,448 protein pairs (228,317 positive, 456,131 negative) and PiNUI-yeast contains 157,035 pairs (53,037 positive, 103,998 negative). Both datasets are split randomly into train/validation/test sets with a 60/20/20 ratio; Table 5 summarizes these statistics.

Table 6 reports Test AUROC and AUPRC results across competing methods. The unsupervised ESM2 baseline, i.e., ESM2-NoFT, performs poorly on both datasets, highlighting the importance of fine-tuning for PPI prediction. RaftPPI achieves the best performance on both datasets. These results are consistent with our findings on the seven-dataset benchmark discussed in § 4.2 , demonstrating that the proposed residue-level interaction modeling and adaptive negative weighting in RaftPPI generalize well to larger-scale PPI datasets. PLM-Interact and ESM2-MLP appear to also have strong

| Dataset | Species | Train | | | Val | | | Test | | | Total |
|---|---|---|---|---|---|---|---|---|---|---|---|
| | | Pos | Neg | Total | Pos | Neg | Total | Pos | Neg | Total | |
| PiNUI-human | Human | 136,812 | 273,858 | 410,670 | 45,684 | 91,205 | 136,889 | 45,821 | 91,068 | 136,889 | 684,448 |
| PiNUI-yeast | Yeast | 31,797 | 62,424 | 94,221 | 10,569 | 20,838 | 31,407 | 10,671 | 20,736 | 31,407 | 157,035 |

Table 5: Dataset statistics for PiNUI-human and PiNUI-yeast.

| Model | PiNUI-human | | PiNUI-yeast | |
|---|---|---|---|---|
| | AUROC | AUPRC | AUROC | AUPRC |
| ESM2-NoFT | $59.14 \pm 0.00$ | $42.45 \pm 0.00$ | $51.90 \pm 0.00$ | $38.74 \pm 0.00$ |
| ESM2-MLP | $75.37 \pm 0.66$ | $61.97 \pm 1.03$ | $77.52 \pm 0.76$ | $63.01 \pm 0.81$ |
| PLM-Interact | $76.04 \pm 0.18$ | $62.71 \pm 0.35$ | $76.77 \pm 0.96$ | $61.97 \pm 1.71$ |
| TUnA | $64.53 \pm 0.34$ | $47.31 \pm 0.55$ | $69.79 \pm 1.08$ | $54.29 \pm 0.67$ |
| RaftPPI-P | $73.61 \pm 0.31$ | $61.06 \pm 0.31$ | $72.53 \pm 0.62$ | $58.64 \pm 0.67$ |
| RaftPPI | $\mathbf{77.92 \pm 0.39}$ | $\mathbf{69.33 \pm 0.34}$ | $\mathbf{77.87 \pm 0.41}$ | $\mathbf{69.24 \pm 0.47}$ |

Table 6: Test AUROC and AUPRC performance (%) on PiNUI datasets.

performance. Besides, the improvements of RaftPPI v.s. RaftPPI-P (protein-level interaction only) further demonstrate the effectiveness of our design that considers residue-level interaction.

## C  USE OF LARGE LANGUAGE MODELS

We used a large language model to help polish the writing. We take full responsibility for all content in this paper.

