# OpenReview forum: "Fast Proteome-Scale Protein Interaction Retrieval via Residue-Level Factorization"
_ICLR.cc/2026/Conference — ICLR 2026 Poster_

### Official Review · Reviewer_DB9S · 2025-10-27

**Soundness:** 2
**Presentation:** 3
**Contribution:** 2
**Rating:** 4
**Confidence:** 4

**Summary:**

This work proposes RaftPPI, which models PPI by approximating residue-level interactions while enabling scalable protein retrieval. RaftPPI computes residue–residue scores using a Gaussian kernel and aggregates them into a protein-level score. The non-linear kernel is efficiently approximated via random Fourier features, and pooling is implemented with a low-rank factorized attention mechanism that allows linear-time approximation during inference. Each protein is encoded once into an indexable embedding suitable for ANN search, enabling retrieval of likely interactors while preserving residue-level interactions and avoiding explicit per-pair computation.

**Strengths:**

This work transforms nonlinear residue-level interaction modeling into a factorizable embedding form, substantially enhancing the scalability of proteome-level retrieval. It demonstrates outstanding computational efficiency, achieving an acceleration from GPU-months to minutes.

**Weaknesses:**

1.	The results in Table 2 show no significant improvement, with RaftPPI surpassing PLM-Interact by only 0.15 on the Average metric.
2.	RaftPPI, short for Residue-interaction Approximation with Fourier FeaTures, only validates the effectiveness of the residue-interaction approximation in the ablation studies, lacking evidence for the contribution of the Fourier features.
3.	The paper lacks a discussion on limitations and future research directions, which would help readers gain a more comprehensive understanding of the work.

**Questions:**

Please refer to the above "weakness" part.

---

> ### Author Response · Authors · 2025-11-24
> **Rebuttal by Authors**
>
> We thank the reviewer for the feedbacks. In the rebuttal period, we updated the paper based on them by adding the limitations and future work in Section 5. Below we would like to comment on the identified weaknesses.
>
> ### W1. Magnitude of improvements
> We admit that the improvement of PPI classification in terms of AUROC is not significant, where RaftPPI (75.29) matches PLM-Interact (75.14). However, we would like to emphasize that the main focus of our paper is to provide an efficient and effective solution for PPI retrieval at proteome scale, identifying candidate PPIs requires evaluating nearly all possible protein pairs.
>
> RaftPPI outperforms PLM-Interact at the PPI retrieval task at a large margin. Specifically, as shown the retrieval heatmap (Figure 2) and retrieval-efficiency table (Table 2), RaftPPI reaches an average Recall@20% of 48.33%, **+17.52% improvements** versus PLM-Interact’s 30.81%, while cutting full human-proteome screening from GPU-months to minutes (**over $30{,}000\\times$ speedup**).
>
> ### W2. Evidence for SORF / Fourier features
> The ablation section removes the kernel approximation (“Raft-WoSORF”) and observes drops in both Recall@20% and AUROC, showing that the Gaussian kernel is essential. The bandwidth sweep selects a well-calibrated $\\hat{\sigma}$, and the RFF dimension sweep shows that varying $d'$ leaves performance stable, indicating that SORF provides an effective, well-conditioned approximation rather than noise.
>
> We also find that the SORF kernel bandwidth $\hat{\sigma}$ is particularly important compared to other hyperparameters. For example, on the human dataset, $\hat{\sigma} = 0.5$ achieves 73.1 AUROC, while a poorly tuned bandwidth leads to a substantial drop in AUROC, highlighting that the SORF design and its calibration play a central role in our method.
>
> ### W3. Limitations and future work
> We have included an explicit “Limitations & Future Work” paragraph in the conclusion. For completeness, we briefly summarize them here.
>
> **Limitations**: First, RaftPPI trades exact residue–residue modeling for efficiency via kernel approximation and rank-$r$ attention, which may under-represent subtle allosteric effects or conformational rearrangements at complex interfaces. Second, the model operates in a sequence-only regime without structural templates and is trained on pairwise labels rather than complex-level structural supervision, so it does not directly observe ground-truth residue contact maps and might inherit dataset biases (e.g., assay type, species coverage, interaction density).
>
> **Future Works**: First, we identify structure-aware pretraining as a promising direction, for example on data with residue contact maps or interface distances, so that the kernel and attention can learn more physically grounded interaction patterns while preserving a factorizable retrieval head. In addition, we plan to explore structure-backed retrieval, where RaftPPI proposes candidates that are subsequently refined or rescored by structure prediction models such as AlphaFold3, and to extend our framework to condition-specific PPIs and larger cross-species proteomes with dynamically updatable indices.

---

### Official Review · Reviewer_3PC9 · 2025-10-30

**Soundness:** 3
**Presentation:** 3
**Contribution:** 2
**Rating:** 6
**Confidence:** 3

**Summary:**

SummaryThis paper proposes RaftPPI, a method to speed up proteome-scale protein-protein interaction (PPI) screening. The main problem it addresses is that existing sequence-based models are too slow for an all-against-all search, often requiring $\mathcal{O}(N^{2}L^{2})$ computation. RaftPPI approximates a residue-level interaction model with a factorizable one. It models residue interactions with a Gaussian kernel, approximates this kernel using random Fourier features (SORF), and uses a low-rank (rank-1) attention mechanism to pool these features. This allows each protein to be encoded into a single embedding. The interaction score is then just a dot product of these embeddings, which enables fast retrieval using Approximate Nearest Neighbor (ANN) search. The authors also use an adaptive negative weighting loss to focus on harder negative examples during training. The method is shown to be much faster than prior work, reducing screening time from months to minutes on the human proteome, while achieving strong performance on several classification and retrieval benchmarks.

**Strengths:**

- The primary strength is the massive, practical speedup. Reducing a computation from 4.9 GPU-months to 6 GPU-minutes is a game-changer for the field and makes routine proteome-scale screening a reality.
- The experimental setup is rigorous. By using 7 datasets that are explicitly controlled for sequence similarity and degree bias, the authors provide a robust and trustworthy evaluation of their model's performance.
- The paper is very clear, especially Figure 1, which provides an excellent intuitive explanation of how the factorization is achieved and why it leads to a speedup.
- The ablation study confirms that the different components of the model (the kernel, the attention, and the loss function) all positively contribute to the final performance.

**Weaknesses:**

1. The main weakness is the limited novelty of the individual components. The work is a clever integration of existing techniques: PLM embeddings, kernel approximation with Random Fourier Features (specifically SORF), and low-rank (rank-1) attention. This makes the contribution feel more incremental and engineering-focused rather than a fundamental breakthrough.
2. The use of a rank-1 approximation for the attention mechanism ($r=1)$ is a very strong simplification. It's surprising this works so well and it's not well-justified beyond just stating it "achieves strong performance". This could be a significant limitation, as a rank-1 matrix is unlikely to capture complex residue-residue interaction patterns.
3. The justification for using the ESM2-8M model is weak. The appendix (B.1) bases this choice on an unsupervised scaling analysis of [CLS] token dot products. This is not a good proxy for the full, finetuned RaftPPI model, which uses all residue embeddings in a complex, kernelized way. It's very possible the model's performance is being left on the table by not using a larger, more powerful PLM.
4. There is no discussion of how the Gaussian kernel bandwidth $\hat{\sigma}$ was selected. This is a critical hyperparameter for kernel methods, and its tuning (or lack thereof) could significantly impact performance.

**Questions:**

1. The use of a rank-1 ($r=1$) approximation for the attention mechanismseems like a major simplification. Did you experiment with higher ranks (e.g., $r=4, 8, 16$)? What is the performance-vs-computation trade-off here?
2. How was the Gaussian kernel bandwidth $\hat{\sigma}^{2}$ selected? Can you provide a sensitivity analysis for this hyperparameter?
3. The justification for using the small ESM2-8M model rests on an unsupervised analysis of [CLS] token dot products. Why should this analysis apply to the full RaftPPI model, which is finetuned and uses all residue embeddings in a kernelized way? Have you tried finetuning RaftPPI with a larger PLM (e.g., ESM2-35M) to see if performance improves?
4. The adaptive negative weighting loss is one of several ways to handle hard negatives. How does it compare, both in performance and simplicity, to a more standard, well-tuned focal loss?

---

> ### Author Response · Authors · 2025-11-24
> **Rebuttal by Authors (1/2)**
>
> We thank the reviewer for noting the clear presentation and the substantial runtime savings. Below we address each weakness and question with the new evidence added in the revision.
>
> ### W1. Novelty of the overall contribution
>
> We would like to clarify that RaftPPI goes beyond a straightforward combination of existing components by (i) recasting the Pred&Agg residue-interaction pipeline as a Gaussian-kernel model, (ii) introducing a structured random-feature and rank-one attention factorization that yields single-protein embeddings amenable to ANN retrieval, and (iii) demonstrating that this design enables, for the first time to our knowledge, routine proteome-scale PPI screening with state-of-the-art accuracy.”
>
>
> ### W2. Rank-one attention is too simplified (Q1)
> Empirically, we observe that rank-one attention is sufficient to model residue-level interactions. Specifically, we conduct ablation experiments on attention rank $r\\in\\{1,2,4,8,16,32\\}$. As shown in the table below, rank $r=1$ already achieves the best AUROC and Recall@20%, while $r=2$ slightly improves the very top of the ranking (Recall@1/3/5%) without improving AUROC; larger ranks substantially hurt all metrics despite their higher capacity and $r$-fold larger embedding dimension.
>
> | Attention rank $r$ | AUROC (%) | Recall@1% | Recall@3% | Recall@5% | Recall@10% | Recall@20% |
> | ------------------ | --------- | --------- | --------- | --------- | ---------- | ---------- |
> | 1                  | 75.29     | 10.89     | 18.19     | 23.31     | 33.95      | 48.33      |
> | 2                  | 74.56     | 11.33     | 18.89     | 24.28     | 33.72      | 47.24      |
> | 4                  | 69.49     | 10.23     | 17.12     | 22.10     | 31.47      | 43.89      |
> | 8                  | 66.13     | 8.00      | 13.95     | 18.65     | 27.84      | 40.39      |
> | 16                 | 63.86     | 7.08      | 12.86     | 17.16     | 25.50      | 37.91      |
> | 32                 | 64.71     | 7.46      | 12.98     | 17.37     | 25.76      | 38.19      |
>
> Since rank-one attention is sufficiently effective while achieving the best efficiency compared to higher ranks, we keep $r=1$ as the default.
>
> ### W3. Backbone scale (Q3)
>
> We admit that the unsupervised scaling performance might not be a good proxy for the full, finetuned RaftPPI model. Here we extend the scaling study with a finetuned RaftPPI-35M run. The table below shows a small improvement over the 8M model:
>
> | Method      | AUROC | Recall@20% |
> | ----------- | ----- | ---------- |
> | RaftPPI-8M  | 75.29 | 48.33      |
> | RaftPPI-35M | 75.33 | 49.06      |
>
> Please kindly note that we select ESM-8M as the backbone due to the computational intractability of performing large-scale experiments for non-factorizable models. For example, PLM-Interact would require 4.9 GPU-months to search the human interactome (see Table 3).
>
>
> ### W4. Gaussian kernel bandwidth (Q2)
>
> The Gaussian kernel bandwidth $\\hat{\sigma}$ in Eq. 3 controls how strictly residue-level interactions are determined: smaller $\\hat{\sigma}$ values conﬁne each residue to interact only with very close neighbors (capturing sharp, local interfaces), whereas larger $\\hat{\sigma}$ allows larger interaction scores over a broader neighborhood. We select $\\hat{\sigma}=0.5$ for all the datasets based on performance on validation sets. Here we provide a detailed parameter sensitivity analysis. The table below sweeps $\\hat{\sigma}\\in\\{0.125,0.25,0.5,1,2,4,8\\}$; the mean PPI classification and retrieval performance (averaged over human or yeast benchmarks) is:
>
> | Kernel Bandwidth $\hat{\sigma}$ | Human AUROC | Human Recall@20% | Yeast AUROC | Yeast Recall@20% |
> | ------------------------------- | ----------- | ---------------- | ----------- | ---------------- |
> | 0.125                           | 70.22       | 40.60            | 78.73       | 46.61            |
> | 0.25                            | 72.22       | 45.01            | 80.37       | 53.54            |
> | 0.5                             | 73.14       | 47.14            | 80.88       | 51.91            |
> | 1.0                             | 70.68       | 45.17            | 79.53       | 51.35            |
> | 2.0                             | 67.85       | 42.75            | 77.02       | 48.21            |
> | 4.0                             | 62.26       | 37.35            | 73.53       | 48.07            |
> | 8.0                             | 56.48       | 31.01            | 69.75       | 44.24            |
>
> Generally, $\hat{\sigma}=0.5$ yields the best performance for three out of four metrics. All metrics degrade for very small or very large bandwidths, indicating that RaftPPI benefits from moderately localized kernels and is well-behaved in the neighborhood of our default $\hat{\sigma}=0.5$ (see Appendix B.2 for a detailed visualization and discussion).

---

> > ### Author Response · Authors · 2025-11-24
> > **Rebuttal by Authors (2/2)**
> >
> > ### Response to Questions
> >
> > For Q1, please refer to our response to W2.
> >
> > For Q2, please refer to our response to W4.
> >
> > For Q3, please refer to our response to W3.
> >
> > Response to Q4 Comparison against Focal Loss:
> >
> > Focal Loss was originally proposed for addressing the problem of class imbalance in standard classification settings, where it modulates and balances positive and negative samples using the (α) parameter and further obtains different sample weights based on the $(1 - p)^{\\gamma}$ term. Our adaptive negative sample weighting, inspired by [1], comes from the knowledge graph reasoning (KGR) domain, which typically features (1) sparse ground truth pairs and (2) ranking-style objectives where the observed positive pair is encouraged to have a higher score compared with many sampled (unreliable) negative pairs. In this setting, it is natural to only reweight negative samples based on their relative hardness within a batch. We view our PPI retrieval scenario as more closely aligned with the KGR setting (as PPI retrieval aims to find the interacting pairs in the proteome and negative PPIs are often sampled) than with conventional CV classification.
> >
> > Performance-wise, we provide additional experiments on focal loss. The table below reports the seven-dataset average performances of PPI classification and retrieval. We can observe that both adaptive negative loss and Focal Loss improve over standard BCE, indicating that PPI samples indeed have different effective quality, and our adaptive negative loss is slightly better on average.
> >
> > | Model                   | Test AUROC | Recall@20% |
> > | ----------------------- | ---------- | ---------- |
> > | RaftPPI-AdaptiveNegLoss | 75.29      | 48.33      |
> > | RaftPPI-FocalLoss       | 74.16      | 47.60      |
> > | RaftPPI-BCE             | 72.24      | 45.05      |
> >
> >
> >
> > ### Reference
> >
> > [1] Zhiqing Sun, Zhi-Hong Deng, Jian-Yun Nie, Jian Tang. “RotatE: Knowledge Graph Embedding by Relational Rotation in Complex Space” ICLR 2019.

---

> > > ### Comment · Reviewer_3PC9 · 2025-11-27
> > >
> > > Thank you for your rebuttal. I'll keep my score.

---

### Official Review · Reviewer_nANi · 2025-10-31

**Soundness:** 2
**Presentation:** 3
**Contribution:** 3
**Rating:** 4
**Confidence:** 2

**Summary:**

This paper introduces RaftPPI, a method that is a computationally efficient framework which is designed to quickly and accurately predict protein-protein interactions (PPIs) on massive protein datasets. The central problem it solves is that existing high-accuracy models are too slow, requiring months of computation to screen an entire dataset. RaftPPI approximates complex, residue-level (amino acid) interactions using an efficient method that encodes each protein into a single, compact, and indexable embedding. This "factorizable" approach, which uses a Gaussian kernel approximated by random Fourier features, allows the model to replace exhaustive pairwise comparisons with a fast nearest-neighbor search. As a result, RaftPPI can bring multiple folds of speedup, while achieving superior accuracy in both PPI classification and retrieval.

**Strengths:**

Here are some strengths of the paper:

1. $\textbf{Computational Efficiency}$

The paper's main strength is that it directly tackles the most significant bottleneck in protein-protein interaction, which is computational cost. Traditional high-accuracy models are too slow, taking "GPU-months" as described in the paper. This paper proposes a method that reduces this significantly by multiple folds. This makes large-scale interaction screening very practical.

2. $\textbf{Proposed Method}$

The proposed method solves the efficiency problem in a clever way without compromising the performance of protein-protein interaction performance. Instead of comparing every residue on one protein to every residue on another protein for all possible pairs ($O(N^2L^2)$), it creates a single, compact embedding for each protein which allows the problem to be re-framed as an efficient nearest-neighbor search with $O(NL^2)$ complexity.

3. $\textbf{Superior Results }$

The authors show that in average their method performs better than baselines in different datasets.

**Weaknesses:**

Here are some weakness/questions:

1. $\textbf{Additional datasets}$

How will the proposed method perform on datasets like the one shown in [1] which is a dataset proposed for protein-protein interaction for ML pipeline?

2. $\textbf{Comparison with bigger models}$

Is it possible that your model misses subtle, complex, or non-linear interactions that a full, slower model (like AlphaFold) might capture? Is there a way to compare this quantitatively?

3. $\textbf{Limitations of the work}$

I don't see any place where the limitation of the proposed work being mentioned. What are some discussions/limitations of this approach? It would be beneficial to include those in the paper or appendix as well.

4. $\textbf{Interpretability}$

How interpretable is your approach? Can it be used with works that require interpretability?


References

1. PiNUI: A Dataset of Protein-Protein Interactions for Machine Learning; Geoffroy Dubourg-Felonneau and Eyal Akiva and Daniel Wesego and Ranjani Varadan; NeurIPS 2023 Workshop on New Frontiers of AI for Drug Discovery and Development; 2023

**Questions:**

Please look at the limitation section

---

> ### Author Response · Authors · 2025-11-24
> **Rebuttal by Authors (1/2)**
>
> We thank the reviewer for the constructive feedbacks. Below we would like to comment on the identified weaknesses.
>
> ### W1. PiNUI datasets
> We appreciate the suggestion to evaluate on the large-scale PiNUI datasets and have added these results to Appendix B.6 of the manuscript. Below we provide the statistics for the PiNUI datasets and performance tables:
>
> | Dataset       | Species | Train Pos | Train Neg | Train Total | Val Pos | Val Neg | Val Total | Test Pos | Test Neg | Test Total | Total   |
> |--------------|---------|-----------|-----------|-------------|---------|---------|-----------|----------|----------|------------|---------|
> | PiNUI-human  | Human   | 136,812   | 273,858   | 410,670     | 45,684  | 91,205  | 136,889   | 45,821   | 91,068   | 136,889    | 684,448 |
> | PiNUI-yeast  | Yeast   | 31,797    | 62,424    | 94,221      | 10,569  | 20,838  | 31,407    | 10,671   | 20,736   | 31,407     | 157,035 |
>
> | Model        | PiNUI-human AUROC | PiNUI-human AUPRC | PiNUI-yeast AUROC | PiNUI-yeast AUPRC |
> |-------------|-------------------|------------------|-------------------|------------------|
> | ESM2-NoFT   | 59.14 ± 0.00      | 42.45 ± 0.00     | 51.90 ± 0.00      | 38.74 ± 0.00     |
> | ESM2-MLP    | 75.37 ± 0.66      | 61.97 ± 1.03     | 77.52 ± 0.76      | 63.01 ± 0.81     |
> | PLM-Interact| 76.04 ± 0.18      | 62.71 ± 0.35     | 76.77 ± 0.96      | 61.97 ± 1.71     |
> | TUnA        | 64.53 ± 0.34      | 47.31 ± 0.55     | 69.79 ± 1.08      | 54.29 ± 0.67     |
> | RaftPPI-P   | 73.61 ± 0.31      | 61.06 ± 0.31     | 72.53 ± 0.62      | 58.64 ± 0.67     |
> | RaftPPI     | 77.92 ± 0.39      | 69.33 ± 0.34     | 77.87 ± 0.41      | 69.24 ± 0.47     |
>
> We observe that these conclusions are consistent with our findings on the seven-dataset benchmark in Section 4.2. In particular, RaftPPI, PLM-Interact, and ESM2-MLP all perform strongly and substantially outperform ESM2-NoFT. RaftPPI achieves the best performance on both AUROC and AUPRC. We also observe substantial gains of RaftPPI over RaftPPI-P (protein-level interaction only), further demonstrating the benefit of explicitly modeling residue-level interactions.
>
> ### W2. Comparison with larger structure-based models
> We acknowledge that complex structure-based models such as AlphaFold2/3 are highly capable for PPI prediction. However, there is no straightforward way to directly compare RaftPPI against these models for the following reasons:
>
> First, the inputs and outputs of the models are fundamentally different. Structure prediction models such as AlphaFold typically require multiple sequence alignments (MSAs) and, when available, structural templates, and they predict full 3D complex structures, whereas RaftPPI is trained on sequence-only PPI classification datasets. It is therefore not possible to train structure prediction directly on our benchmarks for fair comparison.
>
> Second, current structure prediction models are computationally intractable for proteome-level tasks. Structure prediction models are non-factorizable: to obtain scores for all candidate PPIs, one must explicitly evaluate all protein pairs, leading to $O(N^2 L^3)$ complexity ($N$ proteins, average length $L$). In contrast, RaftPPI factorizes the problem via protein embeddings and can retrieve the top-20% candidate interactions for the full human proteome (approximately 20k proteins) in about 6 minutes.
>
> In practice, these models are complementary in proteome-scale PPI retrieval. Sequence-only methods such as RaftPPI are often used as fast filters to select a small subset of promising candidates [1,2], after which structure prediction models are applied to compute structure confidence scores and prioritize pairs for wet-lab validation. We envision our proposed RaftPPI as an efficient and effective alternative to the existing sequence-only filters like DCA.
>
> ### W3. Limitations
> We have included an explicit “Limitations & Future Work” paragraph in the conclusion. For completeness, we briefly summarize the main limitations here. First, RaftPPI trades exact residue–residue modeling for efficiency via kernel approximation and rank-$r$ attention, which may under-represent subtle allosteric effects or conformational rearrangements at complex interfaces. Second, the model operates in a sequence-only regime without structural templates and is trained on pairwise labels rather than complex-level structural supervision, so it does not directly observe ground-truth residue contact maps and might inherit dataset biases (e.g., assay type, species coverage, interaction density).
>
> ### W4. Interpretability
>
> RaftPPI is interpretable via its attention weights and explicit residue–residue interaction scores. In principle, one could compare these residue-pair scores against known complex structures; however, due to limited time in the rebuttal phase, we were not able to include such analyses.

---

> > ### Author Response · Authors · 2025-11-24
> > **Rebuttal by Authors (2/2)**
> >
> > ### References
> > [1] Ian R. Humphreys, Jing Zhang, Minkyung Baek, Yaxi Wang, Aditya Krishnakumar, Jimin Pei, Ivan Anishchenko, Catherine A. Tower, Blake A. Jackson, Thulasi Warrier, Deborah T. Hung, S. Brook Peterson, Joseph D. Mougous, Qian Cong, and David Baker. "Protein interactions in human pathogens revealed through deep learning." Nature Microbiology, 9(10):2642–2652, September 2024.
> >
> > [2] Jing Zhang, Ian R. Humphreys, Jimin Pei, Jinuk Kim, Chulwon Choi, Rongqing Yuan, Jesse Durham, Siqi Liu, Hee-Jung Choi, Minkyung Baek, David Baker, and Qian Cong. "Computing the human interactome." bioRxiv, 2024. doi: 10.1101/2024.10.01.615885.

---

> > > ### Comment · Reviewer_nANi · 2025-11-26
> > > **Follow up**
> > >
> > > Dear Authors,
> > >
> > > Thank you for the replies. Based on the discussion and the updated results in the paper, I will adjust my score accordingly.

---

### Official Review · Reviewer_6gN1 · 2025-11-01

**Soundness:** 2
**Presentation:** 3
**Contribution:** 3
**Rating:** 6
**Confidence:** 3

**Summary:**

The paper introduces RaftPPI, a sequence‑only framework that preserves residue‑level fidelity yet enables proteome‑scale retrieval. The method (i) models residue–residue scores by a Gaussian kernel on PLM residue embeddings and (ii) uses low‑rank (separable) attention so that the aggregation over residue pairs becomes a dot product between per‑protein embeddings after a Structured Orthogonal Random Features (SORF) transform; this permits HNSW‑based ANN search using one‑time per‑protein encodings. The training objective applies adaptive negative weighting to emphasize hard negatives. On seven-degree and homology‑controlled datasets, RaftPPI attains the best mean AUROC and best mean Recall@20%, while reducing whole‑human‑proteome retrieval from ~148.5 A100 GPU‑days for PLM‑Interact to minutes using a single GPU.

**Strengths:**

- **(S1) Effective methods.** Modeling residue interactions via a Gaussian kernel and approximating it with RFF/SORF yields an inner‑product surrogate for the attention‑weighted residue sum. The low‑rank attention gives interpretable residue weights while keeping the index dimension small.

- **(S3) Strong experimental results.** Firstly, the comparison experiment results are impressive. On seven datasets curated to mitigate sequence/degree leakage, RaftPPI achieves the SOTA results, while compressing human‑proteome retrieval from ~148.47 A100 GPU‑days for PLM‑Interact to 241s for Recall@20% after a one‑time encoding. Meanwhile, the paper evaluates against classical and PLM baselines with a shared ESM2‑8M backbone where applicable, runs five seeds, and includes ablations showing that kernelization (SORF), attention pooling, and adaptive negative weighting are each necessary. The Appendix scaling study (Figs. 4–5) justifies the 8M backbone for throughput without sacrificing accuracy.

- **(S3) Clear motivation and professional presentation.** The manuscript is well organized and written in clear language with comprehensive figures and tables. The methodology section is well structured. The technical descriptions are coherent and self-contained.

**Weaknesses:**

- **(W1) Approximation fidelity lacks characterization.** The end‑to‑end error induced by rank‑1 separable attention and finite‑feature RFF is not quantified; no bounds or diagnostics relate the surrogate $<\hat{h}_A, \hat{h}_B>$ to the original attention‑weighted kernel sum. This leaves open whether interfaces requiring multiple spatial patches(multi‑epitope interactions) demand r>1 or larger feature budgets.

- **(W2) Retrieval evaluation underrepresents full proteome use‑cases.** The protocol uses 100 query proteins per dataset rather than all queries in a proteome or cross‑species searches (Fig. 2), so robustness of HNSW hyper‑parameters, index memory/latency vs. N, and failure modes at scale remain unclear. End‑to‑end curves vs. database size or per‑species full‑coverage runs would better support the proteome‑scale claim.

- **(W3) Metrics for extreme imbalance are incomplete.** The paper reports AUROC and Recall@K%, which are informative but insufficient in heavily imbalanced retrieval; AUPRC/average precision, MAP, and NDCG would make performance at the head of the ranking clearer and facilitate comparisons with practical screening budgets.

- **(W4) Limited sensitivity analysis of key design knobs.** The model fixes r=1, d′=2048, and a single bandwidth, and it stops gradients through the negative weights. However, there is no study of how rank r, feature dimension d′, or bandwidth selection trades accuracy, memory, and latency, nor a comparison to allowing gradients through $p_i$. These are central for practitioners who must size the index and allocate compute.

**Questions:**

- **(Q1)** Please detail the negative sampling protocol(s) used in training across datasets (e.g., random, compartment‑aware, topology‑aware) and the ratio of negatives to positives per batch, so others can reproduce the adaptive weighting behavior.

- **(Q2)** Did the authors compare inner‑product vs. cosine retrieval (with/without L2 normalization) in HNSW, and if so, how did this choice affect recall and calibration of scores? A short note or table would guide practitioners.

- **(Q3)** Did the authors try index‑time compression (e.g., product quantization) to reduce memory for whole‑proteome indices, and how does it trade a small loss in recall for a large RAM reduction? It would be valuable for deployment at a larger N.

---

> ### Author Response · Authors · 2025-11-24
> **Rebuttal by Authors (1/3)**
>
> ## Response
>
> Thank you for appreciating our contributions regarding the kernelized factorization, rigorous experiments on seven datasets, and the significant runtime speedup. Below, we address each weakness and question with the new analyses added to the paper revision.
>
> ### W1 — Characterizing approximation fidelity
>
> We thank the suggestion for the pointing out the lack of clarity for approximation techniques, i.e. SORF and low rank attention. Here, we provide a thorough discussion on these techniques and discuss their approximation ability v.s. performance trade-off.
>
> ### 1.1 SORF approximation
>
> **Higher RFF dimension d' theoretically guarantees better approximation**
>
> The SORF approximation approximates the non-linear residue interaction defined in Eq. 3 via random Fourier features. Classical RFF theory [1] for Gaussian kernels shows that the Monte Carlo kernel estimator
>
> $$
> \\hat{k}\_{d^{\\prime}}(x, y)
> := \\phi(x)^{\\top} \\phi(y)
> := \\frac{1}{d^{\\prime}} \\sum\_{j=1}^{d^{\\prime}}
>   \\cos\\left(\\omega\_j^{\\top} x-\\omega\_j^{\\top} y\\right)
> $$
>
> is an unbiased estimator of the true kernel $k(x, y)$, and that its mean-squared error decays as
>
> $$
> \\mathbb{E}\\left[
>   \\left(\\hat{k}\_{d^{\\prime}}(x, y)-k(x, y)\\right)^2
> \\right]
> \\leq \\frac{C}{d^{\\prime}}
> := O\\left(\\frac{1}{d^{\\prime}}\\right)
> $$
>
> for some constant $C>0$. Orthogonal Random Features [2] and their fast variant SORF preserve unbiasedness while further reducing the variance, so that
>
> $$
> \\operatorname{Var}\\left[
>   \\hat{k}\_{d^{\\prime}}^{\\mathrm{SORF}}(x, y)
> \\right]
> \\leq
> \\operatorname{Var}\\left[
>   \\hat{k}\_{d^{\\prime}}^{\\mathrm{RFF}}(x, y)
> \\right]
> := O\\left(\\frac{1}{d^{\\prime}}\\right)
> $$
>
> Therefore, the larger the feature dimension $d^{\\prime}$, the lower the kernel approximation error.
>
> **RaftPPI is not sensitive to RFF dimension**
>
> Empirically, Figure 8 and the table below sweep $d^{\prime} \in\{256,512,1024,2048,4096\}$ (sin/cos features are $2 d^{\prime}$ ); for both human and yeast datasets, AUROC and Recall@20% vary within about 1 point once $d^{\prime}\ge 1024$, indicating that the approximation error is already negligible at our default size and that further increasing $d^{\prime}$ yields diminishing returns.
>
> | $d'$ (RFF dim) | Human-AUROC | Human-Recall@20% | Yeast-AUROC | Yeast-Recall@20% |
> | --- | --- | --- | --- | --- |
> | 256 | 72.20 | 46.21 | 81.07 | 50.38 |
> | 512 | 72.55 | 47.08 | 81.19 | 51.04 |
> | 1024 | 72.82 | 47.13 | 81.00 | 51.59 |
> | 2048 | 73.60 | 47.39 | 81.09 | 50.68 |
> | 4096 | 72.62 | 47.30 | 81.05 | 51.75 |
>
>
> ### 1.2 Attention rank
> We also conduct ablation experiments on attention rank. As shown in table below, rank $r=1$ already achieves the best AUROC and Recall@20%, while $r=2$ slightly improves the very top of the ranking (Recall@1/3/5%) without improving AUROC; larger ranks substantially hurt all metrics despite their higher capacity and $r$-fold larger embedding dimension.
>
> | Attention rank $r$ | AUROC (%) | Recall@1% | Recall@3% | Recall@5% | Recall@10% | Recall@20% |
> | --- | --- | --- | --- | --- | --- | --- |
> | 1 | 75.29 | 10.89 | 18.19 | 23.31 | 33.95 | 48.33 |
> | 2 | 74.56 | 11.33 | 18.89 | 24.28 | 33.72 | 47.24 |
> | 4 | 69.49 | 10.23 | 17.12 | 22.10 | 31.47 | 43.89 |
> | 8 | 66.13 | 8.00 | 13.95 | 18.65 | 27.84 | 40.39 |
> | 16 | 63.86 | 7.08 | 12.86 | 17.16 | 25.50 | 37.91 |
> | 32 | 64.71 | 7.46 | 12.98 | 17.37 | 25.76 | 38.19 |
>
> ### Summary:
>
> Overall, both diagnostics show that the factorizable surrogate already preserves the attention-weighted kernel well: larger $d'$ or $r$ (i.e., tighter approximations) do not neccesarily yield better classification/retrieval performance, while they increase memory/runtime. We therefore argue the current $d'=2048, r=1$ setting offers a fast yet sufficiently faithful approximation.

---

> > ### Author Response · Authors · 2025-11-24
> > **Rebuttal by Authors (2/3)**
> >
> > ### W2 — 100 Query Proteins are Too Few Compared to Full-Proteome
> >
> > We sample 100 queries per dataset so unfactorizable baselines remain computationally tractable—e.g., PLM-Interact would need $12{,}827{,}660$ s ($\approx$148.5 GPU-days) to retrieve top-20% over the full human proteome. Across the 7 datasets this still evaluates 3,902 protein pairs (each query has multiple interactors), so the sampled protocol provides a stable estimate.
> >
> > For factorizable models we additionally run full-proteome retrieval; the table below combines encoding + retrieval time for Recall@20% and compares sampled vs. full Recall@20%. We can see that the sampled-vs-full gap is < 2 percentage points, showing that the 100-query estimate reliably reflects full-proteome recall while keeping unfactorizable baselines feasible.
> >
> > | Type | Model | Retrieve Top-20% time (s) | Recall@20% est. / full proteome (diff%) |
> > | --- | --- | --- | --- |
> > | Unfactorizable | ESM2-MLP | 1,766,576 | 27.77 / NA |
> > | Unfactorizable | TUnA | 3,833,646 | 30.44 / NA |
> > | Unfactorizable | PLM-Interact | 12,827,660 | 30.81 / NA |
> > | Factorizable | ESM2-NoFT | 364 | 42.37 / 41.72 (0.65) |
> > | Factorizable | RaftPPI-P | 241 | 45.73 / 43.83 (1.90) |
> > | Factorizable | RaftPPI | 343 | 48.33 / 47.91 (0.42) |
> >
> > Given the huge performance performance gain (+17.52% Recall@20%) of RaftPPI against the existing SoTA model PLM-Interact, and the relative low estimation error using sampled 100 proteins, it's safe to say that RaftPPI is a much better PPI retrieval model compared with PLM-Interact with a significant speed up ($>30{,}000\\times$).
> >
> >
> > ### W3 — Metrics under extreme imbalance
> > We thank the reviewer for emphasizing evaluation metrics under extreme imbalance. For PPI classification, AUROC and AUPRC are indeed standard; we report AUROC as the primary metric because the recent analysis [3] shows that AUPRC can unduly amplify apparent gains in subpopulations with higher positive rates, which can be misleading when comparing across datasets or rebalancing splits. In newly added experiments on the PiNUI datasets (Appendix B.6, Table 7), we also report AUPRC and observe that it tracks AUROC closely (better AUROC consistently yields better AUPRC), supporting that our conclusions are robust to this choice of metric.
> >
> > For PPI retrieval, our sequence-only models are used as a high-throughput pre-filter to surface a small candidate set for downstream analysis (e.g., structure-based or experimental validation), so the relevant operating point is “how many true interactors are recovered within a fixed budget $K$” rather than the exact global ordering of all candidates. Metrics such as MAP and NDCG require a precise ranking over the entire proteome. For factorizable models queried via HNSW, this would entail recomputing exact scores for every protein pair rather than using approximate ANN neighbors—substantially increasing computation and obscuring the cost advantage we aim to demonstrate.
> >
> > Therefore, we adopt Recall@K% as our primary retrieval metric: it directly measures coverage at a fixed screening budget, is robust under severe class imbalance, and does not depend on exact scores for all non-retrieved proteins.
> >
> > ### W4 — Sensitivity to rank, RFF dimension, bandwidth, and loss
> > **Bandwidth $\\hat{\sigma}$.** The Gaussian kernel bandwidth $\\hat{\sigma}$ in Eq. 3 controls how strictly residue-level interactions are determined: smaller $\\hat{\sigma}$ values conﬁne each residue to interact only with very close neighbors (capturing sharp, local interfaces), whereas larger $\\hat{\sigma}$ allows larger interaction scores over a broader neighborhood. The table below sweep $\\hat{\sigma}\in\{0.125,0.25,0.5,1,2,4,8\}$, the mean PPI classification and retrieval performance (averaged over human or yeast benchmarks) is:
> >
> > | Kernel Bandwidth $\hat{\sigma}$ | Human AUROC | Human Recall@20% | Yeast AUROC | Yeast Recall@20% |
> > | --- | --- | --- | --- | --- |
> > | 0.125 | 70.22 | 40.60 | 78.73 | 46.61 |
> > | 0.25 | 72.22 | 45.01 | 80.37 | 53.54 |
> > | 0.5 | 73.14 | 47.14 | 80.88 | 51.91 |
> > | 1.0 | 70.68 | 45.17 | 79.53 | 51.35 |
> > | 2.0 | 67.85 | 42.75 | 77.02 | 48.21 |
> > | 4.0 | 62.26 | 37.35 | 73.53 | 48.07 |
> > | 8.0 | 56.48 | 31.01 | 69.75 | 44.24 |
> >
> > Genrally, $\hat{\sigma}=0.5$ yields best performance for three out of four metrics. All metrics degrade for very small or very large bandwidths, indicating that RaftPPI benefits from moderately localized kernels and is well-behaved in the neighborhood of our default $\hat{\sigma}=0.5$ (see Appendix B.2 for a detailed visualization and discussion).
> >
> > Attention rank $r$ and RFF dimension $d'$ are analyzed in W1; higher $r$ or larger $d'$ do not improve retrieval and slow indexing. We follow the convention of knowledge graph reasoning [4] to drop gradients. Due to limited time for rebuttal, we do not have time to implement a non-stop-gradient ablation for adaptive negative loss.

---

> > > ### Author Response · Authors · 2025-11-24
> > > **Rebuttal by Authors (3/3)**
> > >
> > > *(Continuing from the previous discussion of W4)*
> > >
> > > In sum, setting kernel band width $\hat{\sigma}=0.5$, attention rank $r=1$ and $d'=2048$ provides a effective performance with efficiency.
> > >
> > > ### Q1. Negative sampling protocol
> > >
> > > We follow the original PPI benchmark processed by [5], where each dataset already provides labeled positive and negative pairs. Consequently, we do **not** perform any additional negative sampling in this paper; the adaptive negative weighting operates on the fixed positives/negatives defined by the benchmark.
> > >
> > > ### Q2 (cosine vs. inner product)
> > >
> > > We thank the reviewer for the constructive suggestion and implement a variant of RaftPPI with $L_2$-normalized protein embeddings, the average results are as follows:
> > >
> > > | Method             | test_auc | recall@1% | recall@3% | recall@5% | recall@10% | recall@20% |
> > > | ------------------ | -------- | --------- | --------- | --------- | ---------- | ---------- |
> > > | RaftPPI-Normalized | 75.74    | 11.28     | 18.87     | 24.35     | 34.61      | 48.80      |
> > > | RaftPPI-WoNorm     | 75.29    | 10.89     | 18.76     | 23.88     | 33.95      | 48.33      |
> > >
> > > Applying normalization improves AUROC from 75.29 to 75.74 and Recall@20% from 48.33 to 48.80, with consistent gains at Recall@1/3/5/10%. We hypothesize that this gain comes from aligning training/inference dot-product scoring with the HNSW index configured for cosine distance—without normalization, norm fluctuations (sequence length, attention distribution, SORF scale) can spuriously favor large-norm embeddings.
> > >
> > > We sincerely appreciate this suggestion, which improves our performance with only a minor one-line code change. For fairness with previously reported results, all other experiments are currently run without normalization; we will rerun the full suite with normalization and update the results in the camera-ready version.
> > >
> > > ### Q3 (index-time compression)
> > >
> > > We agree that index-time compression (e.g., PQ) is important for scaling to even larger proteomes. In the current implementation we index full-precision RaftPPI embeddings in an HNSW backend without any compression so that the reported Recall@K and latency directly reflect the raw representation; at $d'=2048$, the full human-proteome index comfortably fits in RAM and answers top-20% queries in minutes on a single node (Table 3), so memory has not been a bottleneck at our target scale. Because the per-protein embeddings are standard dense float vectors, off-the-shelf index-time compression schemes (e.g., product quantization or float16/binary quantization in the HNSW library) are compatible with RaftPPI and would trade a small loss in Recall@K for a 2–4$\times$ RAM reduction; we see this as an engineering extension for future deployments (e.g., multi-proteome or pan-species indices) rather than a limitation of the current method.
> > >
> > > ### Summary
> > >
> > > We sincerely appreciate your constructive feedbacks, especially the $L_2$-normalization suggestion, which improves our performance with only a minor one-line code change. We hope that our additional experiments on kernel bandwidth, attention rank, RFF dimension, and full-proteome evaluation, together with our responses, address your concerns. We are happy to discuss any further questions you may have.
> > >
> > > ### References
> > >
> > > [1] Ali Rahimi and Benjamin Recht. “Random Features for Large-Scale Kernel Machines.” NeurIPS 2007.
> > >
> > > [2] Felix X. Yu, Ananda Theertha Suresh, Krzysztof Choromanski, Daniel Holtmann-Rice, Sanjiv Kumar. “Orthogonal Random Features.” NeurIPS 2016.
> > >
> > > [3] Matthew B. A. McDermott, Lasse Hyldig Hansen, Haoran Zhang, Giovanni Angelotti, Jack Gallifant. “A Closer Look at AUROC and AUPRC under Class Imbalance” NeurIPS 2024.
> > >
> > > [4] Zhiqing Sun, Zhi-Hong Deng, Jian-Yun Nie, Jian Tang. “RotatE: Knowledge Graph Embedding by Relational Rotation in Complex Space” ICLR 2019.
> > >
> > > [5] Judith Bernett, David B Blumenthal, and Markus List. “Cracking the black box of deep sequence-based protein–protein interaction prediction.” Brieﬁngs in Bioinformatics, 2024.

---

> ### Comment · Reviewer_6gN1 · 2025-11-27
> **Feedback to Rebuttal**
>
> Thank the authors for their thorough responses and for updating a revised manuscript with substantial clarifications. I carefully examined the new revision, cross-checked the updated Sections 3–4, and the detailed responses. Overall, I found my concerns were almost addressed, and the revision improves the clarity. I decided to keep my original positive assessment. Moreover, I suggest that the authors mark all revisions in a consistent color (e.g., blue) to facilitate verification by the AC and reviewers.

---

### Author Response · Authors · 2025-11-24
**General Response by Authors**

We thank the reviewers for their constructive feedback and the time invested in the review process. We appreciate that all reviewers acknowledge both the performance and the speedup achieved by our work. Below, we briefly summarize the main strengths and weaknesses highlighted by the reviewers and provide our general responses.

### Strengths Identified by Reviewers:
- **Effective method design (nANi, 6gN1)**: We propose an effective method to approximate the standard Pred\&Agg pipeline. The design provides a principled way to preserve residue-level modeling while maintaining strong protein–protein interaction performance.
- **Outstanding PPI retrieval speedup (all reviewers)**: Our method achieves significant speedup (over $30{,}000\\times$) for proteome-level PPI retrieval, reducing screening from GPU-months to minutes.
- **Presentation (6gN1, 3PC9)**: Reviewers find the paper clear and well organized.

### Weaknesses Identified by Reviewers
- **Hyper-parameter analysis (6gN1, 3PC9, DB9S)**: Reviewers asked for a more detailed discussion of important hyperparameters. In the updated manuscript, we conduct extensive sensitivity studies. Appendix B now analyzes additional key hyperparameters: Gaussian bandwidth $\\hat{\sigma}$ (B.2), attention rank $r$ (B.3), adaptive temperature $\\tau$ with a focal-loss comparison (B.4), and RFF dimension (B.5). The experiments show that our default setting $\\hat{\sigma}=0.5$, $r=1$, $\\tau=4$, and $d'=2048$ gives the best accuracy/efficiency trade-off; higher ranks and extreme bandwidths do not improve performance, while the RFF dimension is robust across $256$–$4096$.
- **Approximation might be too simple (6gN1, 3PC9)**: Some reviewers are concerned that the SORF-based kernel approximation and low-rank attention might be too simple to capture nuanced residue–residue interactions. Our new sensitivity analyses over the SORF dimension $d'$ and attention rank $r$ show that our chosen approximation is sufficient and that “stronger” approximations (e.g., larger $d'$ or higher $r$) do not necessarily yield better performance.
- **Additional datasets (nANi)**: Per the suggestion of reviewer nANi, we include the PiNUI datasets. Appendix B.6 adds PiNUI-human/yeast results, where RaftPPI attains the best AUROC/AUPRC, demonstrating the generalization beyond the seven controlled splits.
- **Limitations and future work (nANi, DB9S)**: We now provide an explicit discussion of limitations and future work in the conclusion section of the updated paper.

We invite reviewers and the AC to consult the updated paper and detailed point-by-point responses. We sincerely appreciate your feedback and would be happy to address any further questions.

---

### Comment · Area_Chair_VjkF · 2025-11-26

Dear reviewers,

The authors have provided updates and clarifications during the rebuttal, and they are awaiting your follow-up comments.

Please take a moment to review the updates if you haven't already.

Thank you for your efforts.

---

### Meta-Review · Area_Chair_4kij · 2026-01-06

**Summary:**

The paper was reviewed by four reviewers who split in their opinions originally. The authors provided very extensive rebuttals in terms of both more clarifications and more experiments. Overall, majority of the comments have been addressed.

**Reviewer Concerns:**

Major concerns were on experiments, which have been addressed.

**Reviewer Scores:**

The original scores split, but I feel they would improve scores if allowed.

---

### Decision · Program_Chairs · 2026-01-26

Accept (Poster)